# Single-cell profiling reveals differences between human classical adenocarcinoma and mucinous adenocarcinoma

Fang-Jie Hu[1,4], Ying-Jie Li[2,4], Li Zhang[3,4], Deng-Bo Ji[2], Xin-Zhi Liu[2], Yong-Jiu Chen[2], Lin Wang [2✉] & Ai-Wen Wu [2✉]

Colorectal cancer is a highly heterogeneous disease. Most colorectal cancers are classical adenocarcinoma, and mucinous adenocarcinoma is a unique histological subtype that is known to respond poorly to chemoradiotherapy. The difference in prognosis between mucinous adenocarcinoma and classical adenocarcinoma is controversial. Here, to gain insight into the differences between classical adenocarcinoma and mucinous adenocarcinoma, we analyse 7 surgical tumour samples from 4 classical adenocarcinoma and 3 mucinous adenocarcinoma patients by single-cell RNA sequencing. Our results indicate that mucinous adenocarcinoma cancer cells have goblet cell-like properties, and express high levels of goblet cell markers (*REG4*, *SPINK4*, *FCGBP* and *MUC2*) compared to classical adenocarcinoma cancer cells. *TFF3* is essential for the transcriptional regulation of these molecules, and may cooperate with *RPS4X* to eventually lead to the mucinous adenocarcinoma mucus phenotype. The observed molecular characteristics may be critical in the specific biological behavior of mucinous adenocarcinoma.

[1] Department of Gastroenterology, Beijing Chaoyang Hospital, Capital Medical University, Chaoyang District, Beijing 100020, China. [2] Key laboratory of Carcinogenesis and Translational Research (Ministry of Education), Department of Gastrointestinal Surgery III, Peking University Cancer Hospital & Institute, No. 52 Fucheng Rd., Haidian District, Beijing 100142, China. [3] Key Laboratory of Carcinogenesis and Translational Research (Ministry of Education), Department of Pathology, Peking University Cancer Hospital & Institute, Beijing 100142, China. [4] These authors contributed equally: Fang-Jie Hu, Ying-Jie Li, Li Zhang. ✉email: wanglinmd@foxmail.com; wuaw@sina.com

Colorectal cancer (CRC) is the third most common cancer and the second leading cause of cancer-related death worldwide[1]. The most common histologic subtype of CRC is classical adenocarcinoma (AC). Mucinous adenocarcinoma (MC) is a distinct subtype that is characterized by its mucinous components making up at least 50% of the tumour volume[2]. Researches have suggested that 10–20% of CRC patients have the mucinous subtype[3,4]. In terms of clinical pathology, MC is found more common in the proximal colon than in the rectal or distal colon[5,6]. The prevalence of MC is higher than that of AC in female patients and in younger patients[7]. Current treatments for patients with MC are based on the same standard guidelines used for AC. Whether the prognosis of MC is different from that of AC is debated. Some studies have reported that MC is associated with worse survival than AC[8,9], while one study showed that MC and AC have similar survival[10], and yet another study showed better prognosis for MC than for AC[11].

Thus, there is a need to understand the different clinicopathological characteristics and prognoses of these diseases. Several studies have revealed significant differences between MC and AC, suggesting different mechanisms of oncogenesis. Overexpression of the MUC2 protein is one of the most obvious molecular abnormalities that distinguishes MC from AC[12,13]. MC is also associated with a high frequency of microsatellite instability (MSI-H), which is related to Lynch syndrome[14], and with mutations that affect the *RAS-RAF-MEK-ERK* pathway[13]. Moreover, MC features alterations in the expression of *MLH1*, *FHIT*, and *p27* and a decreased rate of *TP53* mutation compared with AC[4,10]. Although specific molecular characteristics of MC have been investigated, all of the studies to data have been based on conventional 'bulk' RNA-sequencing methods, which process a mixture of cells, and average out underlying cell-type-specific differences among transcriptomes. By comparison, single-cell RNA sequencing (scRNA-seq) analyses each cell's gene expression patterns and their cell-to-cell signalling networks. This unbiased characterization allows us to gain clear insights into processes throughout the entire tumour ecosystem, including cell–cell crosstalk through ligand–receptor signalling and mechanisms of intra- and intertumoral heterogeneity[15]. Thus, several studies have characterized the tumour microenvironment (TME) of CRC at single-cell level deeply. Zhang et al. first analysed T-cell subpopulations and illustrated the distinct tumour-infiltrating T lymphocyte landscape of CRC[16]. Stromal cells isolated from the TME were demonstrated to have pervasive genomic alterations that were related to prognosis[17]. Combined analyses of scRNA-seq of TMEs in murine tumour models and CRC identified distinct myeloid populations with differential sensitivity to *CSF1R* blockade and defined concerted immune responses to dendritic cells and T cells relevant for anti-CD40 therapy[18]. However, the unique biological behaviour of MCs has not been well explained, and the mechanism regulating mucus production in these tumours is still unclear.

In the present study, we applied scRNA-seq to analyse the tumour landscape of MC. Our study reveals the goblet cell-like properties of MC cancer cells and might help to clarify different tumour-specific biological features, such as the mucus-richness of MC. Furthermore, the mechanism regulating mucus production is also explained by our findings. Overall, our data provide comprehensive scRNA-seq profile of MC.

## Results

**Establishment of a mucinous adenocarcinoma cell atlas**. We applied scRNA-seq analyses to surgical primary tumour specimens from individuals with nonmucinous CRC (4 patients) and mucinous CRC (3 patients) (Fig. 1a). Nonmucinous and mucinous primary tumour tissues from CRC are referred to as AC and MC, respectively. Detailed clinical and pathological information is provided in Supplementary Table 1. Haematoxylin-eosin (H&E) staining was performed; images of this staining are shown in Fig. 1b. A distinct mucus component was present in MC but not in AC. Following multiple quality control and filtering steps, data from a total of 61,279 cells (33,778 cells from AC; 27,501 cells from MC) were used for further analysis, with more details shown in the supplementary materials (Supplementary Data 1). The median number of detected genes ranged from 874 to 1540 per cell, and the number of detected UMIs (unique molecular indices) ranged from 2780 to 5195 per cell (Supplementary Fig. 1a, b).

Unbiased clustering of the cells identified 9 main clusters in parallel according to their gene profiles and canonical markers, which were visualized through uniform manifold approximation and projection (UMAP) analysis (Fig. 1c–e; Supplementary Table 2). Specifically, the clusters included cancer cells, normal epithelial cells, endothelial cells, pericytes, fibroblasts, plasma cells, myeloid cells, T cells, and B cells. We identified marker genes to separately classify normal epithelial cells and cancer cells (Supplementary Fig. 2a). Subsequent CNVscore analysis confirmed the accuracy of the classification (Supplementary Fig. 2b), such that cancer cells showed a higher CNVscore than normal epithelial cells and myeloid cells. Cancer cells and plasma cells accounted for the majority of cells (Supplementary Data 2). The top 20 differentially expressed genes (DEGs) for the subclusters of 5 of the 9 major clusters are also given (Supplementary Data 3), and more details are shown in the supplementary materials (Supplementary Figs. 3–5) and the following sections.

We noted that immune and stromal cells from different patients clustered together by cell type, and cancer cells exhibited greater heterogeneity and patient-specific expression characteristic (Fig. 1e). The proportions of cancer, stromal, and immune cells varied widely between samples. This variation may be inherent in different tumour phenotypes or related to the locations within the tumour where the samples were taken (Fig. 2a; Supplementary Data 4), suggesting intertumoral heterogeneity as well as consistency among the tumours. We found a higher proportion of epithelial and plasma cells in AC than in MC (although the difference was not significant) (Fig. 2b; Supplementary Data 4).

**Transcriptional heterogeneity of MC cancer cells**. With our attempts to further subdivide the cancer cells, we identified 7 cancer cell subclusters in total, which were visualized through UMAP analysis. Among the subclusters, 4 were related to AC, and 3 were related to MC (Fig. 3a). The gene expression patterns in different clusters of cancer cells were presented in Fig. 3b. The cancer cells exhibit high heterogeneity and patient-specific expression characteristics (Fig. 3c). Among the seven subclusters of cancer cells, cancer cell subcluster 1 specifically expressed high levels of *REG4*, *FCGBP* and *MUC2*, which are considered markers of goblet cells[19,20]. Cancer cell subcluster 2 expressed high levels of *REG1A* and *TM4SF4*. *REG1A* plays a crucial role in alleviating inflammatory injury and maintaining intestinal barrier integrity[21]. *TM4SF4* is likely involved in the epithelial-mesenchymal transition process in CRC[22]. Cancer cell subcluster 2 also expressed high levels of cell proliferation markers (*TOP2A* and *CCND2*) and a goblet cell marker (*CLCA1*). Cancer cell subcluster 3 and cancer cell subcluster 4 were found to be typical malignant epithelial CRC cells, as they specifically expressed *OLFM4*, *CXCL5* and *FABP1*. Cancer cell subcluster 4 showed a strong anti-inflammatory activation state and expressed high levels of immunomodulatory factors including

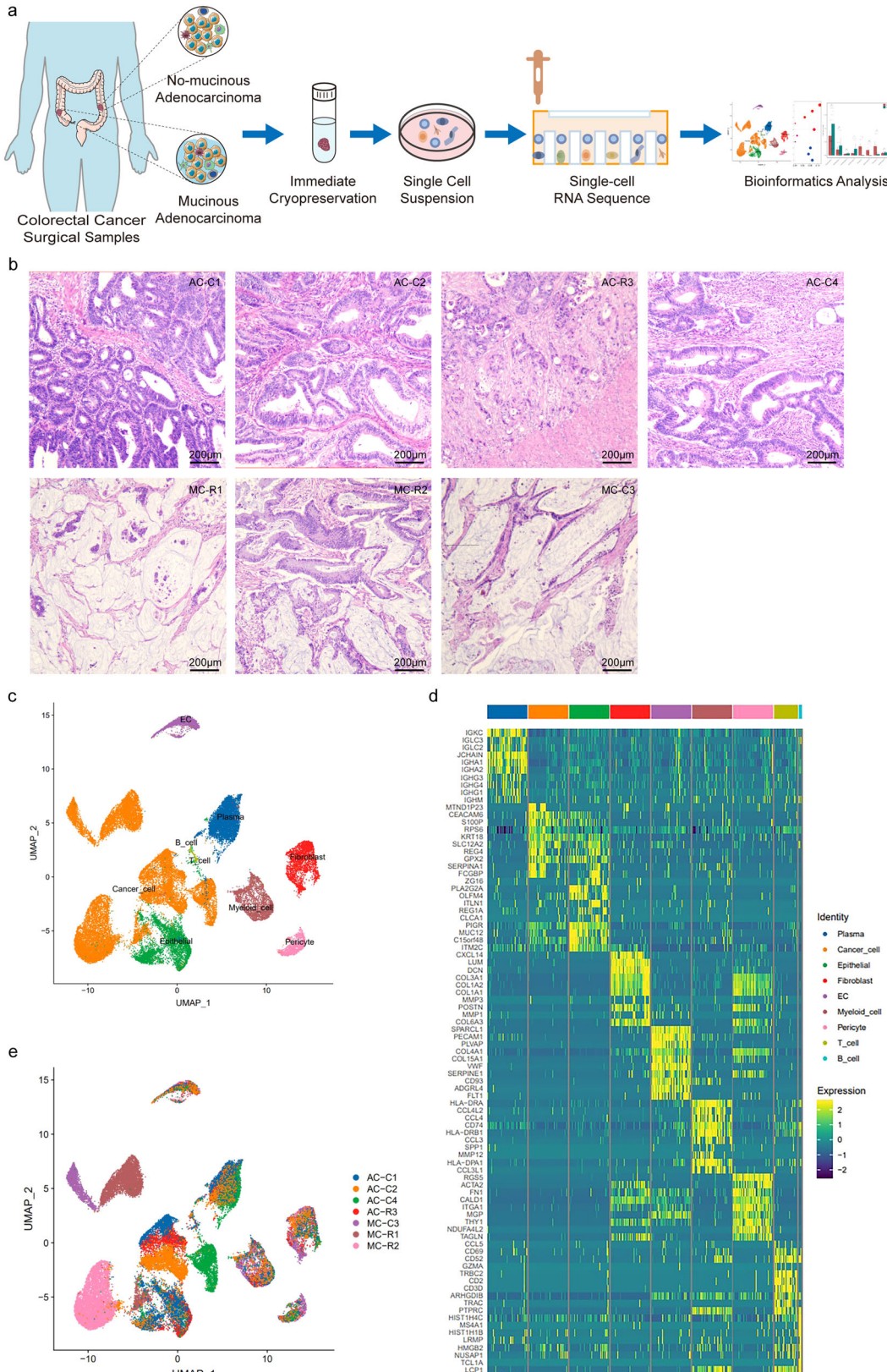

**Fig. 1 Single-cell transcriptomic analysis of CRC lesions. a** Flow diagram of single-cell RNA sequencing. **b** Haematoxylin-eosin (H&E) staining of sequenced samples. Scale bar, 200 μm. **c** Distribution of nine subtypes of cells by UMAP, coloured by cell types. **d** Top ten genes expressed in nine main cell subtypes. Colour key from blue to yellow represents the scaled expression levels of cell type-specific marker genes from low to high. **e** Distribution of seven samples by UMAP, coloured by patients.

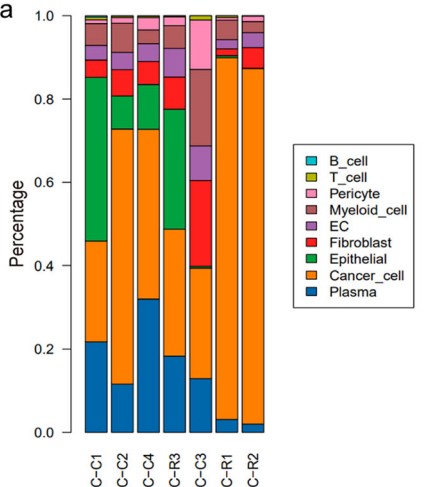
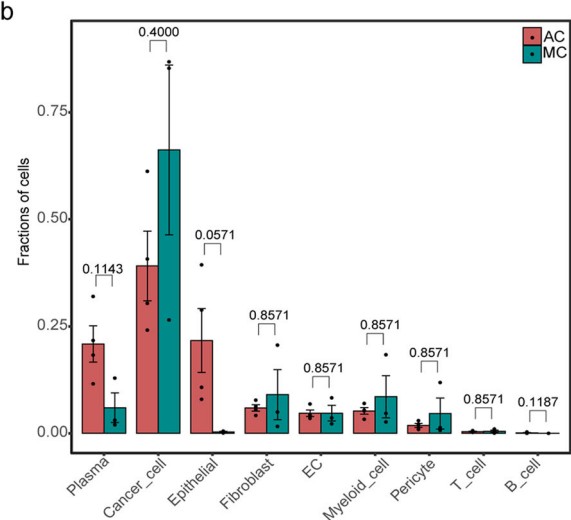

**Fig. 2 Proportion of the nine cell subtypes. a** Proportion of the nine cell subtypes among seven samples. **b** Histogram indicating the proportions of nine main cell clusters in the two groups. The analysis was performed using unpaired two-tailed Wilcoxon rank-sum tests, and statistical significance was set at $p < 0.05$. The error bars represent mean ± std.

*PPBP*, *CCL20*, and *IGLC3*. Cancer cell subcluster 5 specifically expressed the oncogene *CTTN* and the secreted mucin genes *MUC5AC* and *MUC5B*. Cancer cell subcluster 6 specifically expressed *LCP1* and *FN1*, which have been reported to be diagnostic and prognostic markers for CRC[23,24]. Cancer cell subcluster 7 specifically expressed several genes involved in cell growth and proliferation, including *AREG*, *IGF2* and *EREG*. Subsequently, the function of each cluster was identified based on competitive gene set variation analysis (GSVA) (Fig. 3d). Cancer cell subcluster 1 was enriched in *TNFα* signalling via *NF-kB*; cancer cell subclusters 2 and 6 were enriched in the *E2F* and *MYC* signalling pathways; cancer cell subcluster 3 was enriched in the hedgehog, *MYC* and oxidative phosphorylation signalling pathways; cancer cell subcluster 4 was enriched in the *TNFα* via *NF-kB*, *IFN-α* and *KRAS* signalling pathways; cancer cell subcluster 5 was enriched in the epithelial-mesenchymal transition and angiogenesis signalling pathways; and cancer cell subcluster 7 was enriched in the *MYC* and oxidative phosphorylation signalling pathways. These results suggest that CRC is a highly heterogeneous disease. As expected, MC cancer cells expressed high levels of several genes related to mucus formation and stabilization. Interestingly, MC cancer cells expressed high levels of several canonical goblet cell markers.

Next, copy number alterations (CNAs) of cancer cell populations was inferred by using scRNA-seq data (Supplementary Fig. 6a). The inferred CNA profiles from 7 patients indicated both interpatient and intrapatient heterogeneity (Fig. 3e). There were several previously well-defined arm-level changes, including gains of *7p*, *7q*, *8q*, *16p*, *20p* and *20q*[25,26]. Chromosome arm *17p* (which includes *TP53*) was deleted in 57% (4/7) of the lesions. Other significantly deleted chromosome arms were *5q*, *14q* and *22q*. For AC patients, prominent arm-level gains of *20p* and *20q* were found, as were deletions of chromosome *14q*. In contrast, MC patients mostly had *5p* gains and *17p* deletions. Importantly, compared to AC, MC showed specific arm-level changes, including gains of *4p* and *5p* and *19p* deletions. Then, we defined a intratumour heterogeneity score based on CNAs to quantify intratumoral heterogeneity, denoted ITHcna (see Methods for the definitions). We observed various degrees of heterogeneity within tumours (Fig. 3f; Supplementary Data 5). However, there was no significant difference between MC and AC (Supplementary Fig. 6b).

**Unique molecular features of MC cancer cells**. We applied differential expression analyses to identify the specific gene expression patterns of MC cancer cells compared with AC cancer cells. The analysis revealed a total of 744 DEGs, of which 396 were upregulated and 348 were downregulated (Supplementary Data 6). A heatmap of the top 20 DEGs is provided (Fig. 4a). MC expressed high levels of *REG4*, *FCGBP*, *TFF1*, *FAM3D* and *REG1A*, which have been reported to play important roles in protecting the intestinal mucosa and stabilizing the mucous layer[21,27,28]. MC also expressed high levels of *MUC2* and *GNE*, which have been reported to play an important role in mucus secretion and sialic acid synthesis[29,30]. Notably, *REG4*[31], *SPINK4*[20] and *MUC2*[20] are canonical colon goblet cell markers. *FCGBP* is a small intestine goblet cell marker[19]. These results are unsurprising, as colonic mucus is mainly produced by goblet cells. However, the association of MC with goblet cells is very interesting. Subsequently, the function of the two groups was investigated with GSVA (Fig. 4b). MC cancer cells were specifically enriched in signalling pathways related to protein secretion, *TNFα* signalling via *NF-kB* and the early oestrogen response. AC cancer cells were specifically enriched in signalling pathways related to *MYC* and oxidative phosphorylation. GO and KEGG enrichment analyses showed similar results (Supplementary Fig. 7a–d). GO analysis of the genes upregulated in the MC group was performed to determine the main BP terms (glycoprotein biosynthetic process, glycosylation, and intrinsic apoptotic signalling pathway), CC terms (focal adhesion and cell−substrate junction), and MF terms (DNA binding transcription factor binding and unfolded protein binding) (Supplementary Fig. 7a). KEGG enrichment analysis indicated enrichment of the terms "protein processing in the endoplasmic reticulum", "oestrogen signalling pathway" and "antigen processing and presentation" (Supplementary Fig. 7b). The term "biosynthesis of mucin-type O-glycans pathway" was also enriched. In contrast, AC expressed high levels of *FABP1* and *OLFM4*, which are markers of enterocytes and intestinal stem cells[19]. In addition, AC expressed high levels of *PPBP* and *CXCL5*, which are involved in the activation of neutrophils[32]. AC also expressed high levels of *IGHA1*, *IGHA2*, *IGLC3*, and *IGKC*, which play important roles in the activation of B cells. GO enrichment analysis was performed to determine the main BP terms (oxidative phosphorylation and transcription initiation), CC terms (mitochondrial inner membrane and

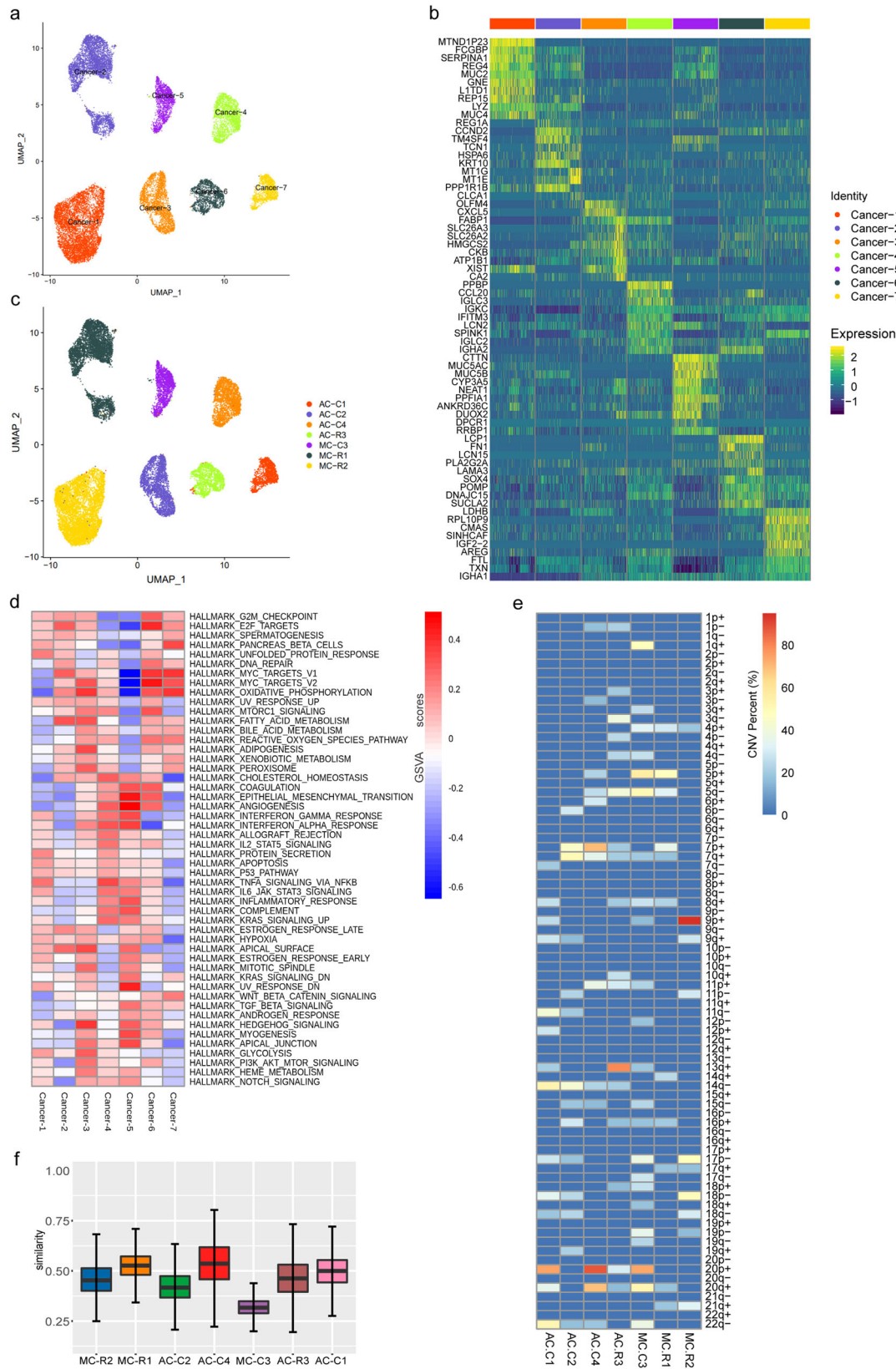

ribosome), and MF terms (structural composition of ribosome) (Supplementary Fig. 7c). KEGG enrichment analysis indicated enrichment of the terms "ribosome" and "oxidative phosphorylation" (Supplementary Fig. 7d).

To further define the phenotypes of MC and AC cancer cells, an unsupervised algorithm, Hotspot analysis was applied (Fig. 4c).

Subsequently, the modular genes were classified as MC or AC signature genes by Jaccard Similarity analysis. Modules 2, 5, 6, and 8 were genetically similar to MC, while modules 4, 10, 13, and 15 were genetically similar to AC (Fig. 4d). Detailed genes for each module are shown in supplementary data 7. The results indicate a strong similarity between module 5 and the phenotype

**Fig. 3 Intertumour and intratumour heterogeneity of cancer cells. a** UMAP visualization of cancer cell clusters. **b** Top ten genes expressed in seven cancer cell subclusters. Colour key from blue to yellow represents the scaled expression levels of cell type-specific marker genes from low to high. **c** Distribution of seven samples based on UMAP analysis; data are coloured by patient. **d** Cluster heatmap of GSVA of 50 hallmark genes from MSigDB among the seven CRC cell subclusters. Colour key from blue to red represents the *GSVA scores* from low to high. **e** CNV profiles of the CRC cells from the seven CRC samples inferred from inferCNV analysis. The CNV levels are categorized by the chromosome arm, where "+" indicates genomic amplifications and "-" indicates genomic deletions in single cells. Colour key from blue to red indicates the percentage of single cells from each individual sample with CNV events from low to high. **f** ITH$_{CNA}$ analysis of the cancer cells from the seven CRC samples. The lower hinge, middle line, and upper hinger of boxplots represented the first, second, and third quartiles of the distributions. The upper and lower whiskers corresponded to the largest and smallest data points within the 1.5 interquartile range.

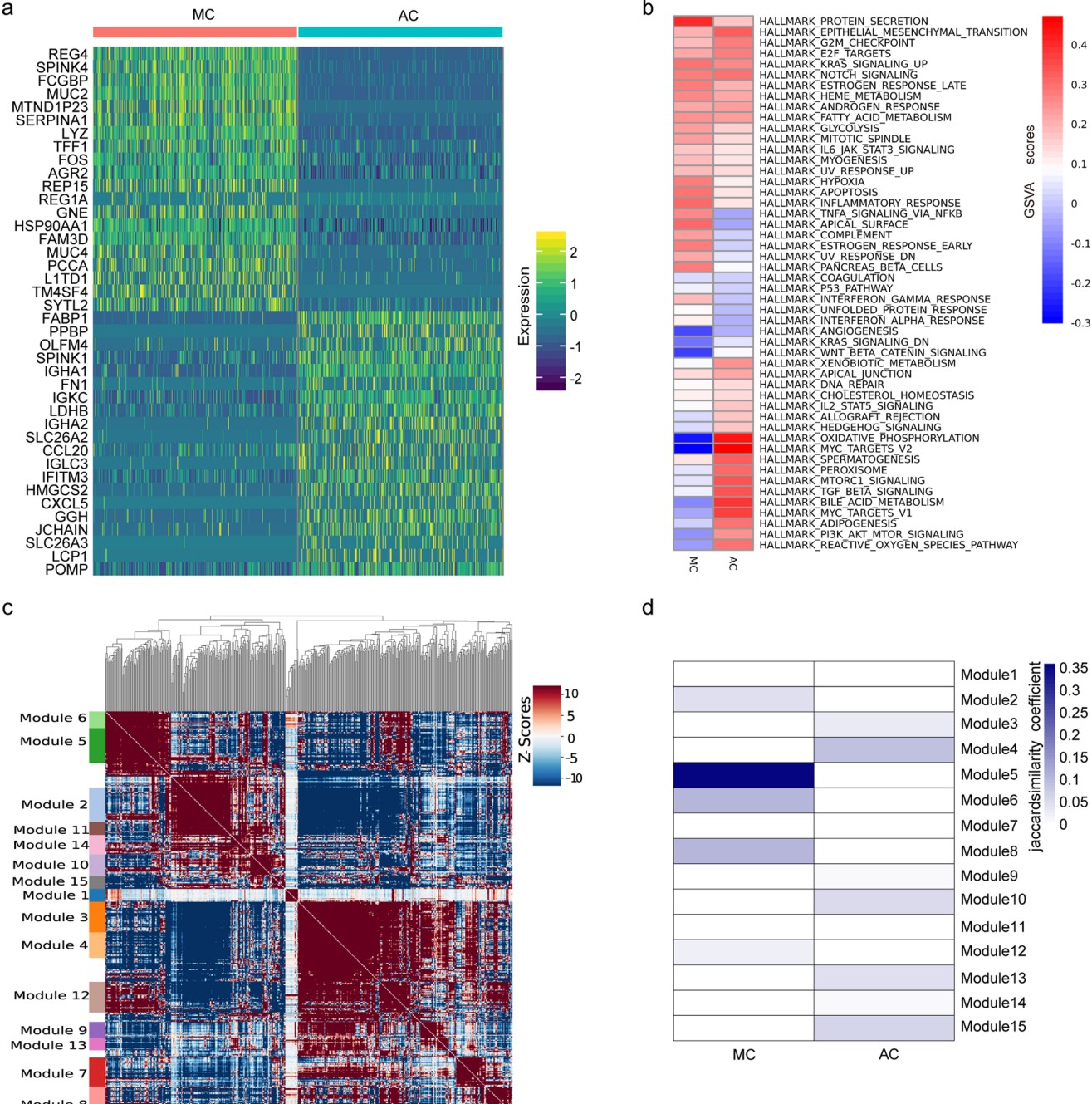

**Fig. 4 Specific molecular characteristics of MC cancer cells. a** Top 20 DEGs between MC and AC. Colour key from blue to yellow represents the scaled expression levels of DEGs from low to high. **b** Heatmap of GSVA of the 50 hallmark gene sets in the MSigDB database between the two groups. Colour key from blue to red represents the GSVA *scores* from low to high. **c** Heatmap of gene correlations, with colour shades representing the *Z score* values. The notes on the left indicate the gene sets. **d** Heatmap of similarity between cell types and gene sets, with colour shades representing the *Jaccard similarity coefficient*.

of MC cancer cells. Consistent with the previous DEG results, goblet marker genes, including *REG4*, *FCGBP*, *SPINK4* and *MUC2*, belong to module 5. To further test our findings, 9 pairs of paired MC tissues (cancer and normal) and 9 pairs of paired AC tissues (cancer and normal) were collected for immunohisto-chemical (IHC) analysis (Fig. 5a). The IHC score was used to determine the level of marker expression (see methods) (Fig. 5b) and detailed information was shown in Supplementary Data 8. There was no difference in the expression of CEACAM6 between MC and AC cancer tissues and CEACAM6 was not expressed in normal tissues. Importantly, REG4, FCGBP, SPINK4 and MUC2 were significantly overexpressed in both MC cancer tissues and normal tissues compared to AC cancer tissues. TCGA survival analysis revealed that patients with CRC expressing high levels of *REG4*, *SPINK4*, *MUC2*, *REP15*, *FAM3D*, *HMGCS2* and *SLC26A3* experienced prolonged OS (Fig. 5c). In summary, our results indicate that *REG4*, *FCGBP*, *SPINK4* and *MUC2* are key genes for the specific phenotype of MC and may improve the prognosis of MC.

Zhang et al. analysed 41 colorectal cancer cell lines by RNA-seq in 2015[33]. To classify whether these cell lines belong to MC or AC, we downloaded RNA-seq data for the 41 colorectal cancer cell lines and evaluated the levels of the genes belonging to MC or AC modules. The expression of module genes for all cell lines is shown (Fig. 6a). The expression levels of genes in the MC or AC groups were calculated to compare the phenotypic tendencies of 41 cell lines (Supplementary Fig. 8). None of the 41 cell lines showed a statistically significant predisposition to the MC or AC phenotype. Notably, HT-29 and LS180 showed the strongest MC propensity of all the cell lines ($p = 0.055$ and $p = 0.093$, respectively) (Fig. 6b; Supplementary Data 9). The expression of MC marker genes (*REG4*, *FCGBP*, *SPINK4*, *MUC2*) was also investigated, and cell lines KM.12, LS.180, CL.34, HCC1263, SK.CO.1, CL.40, HT.115 and LoVo showed MC characteristics (Fig. 6c).

**Trajectory of epithelial cells in MC**. CRC can originate from intestinal stem cells, transit amplifying cells and differentiated cells[34]. Our previous results indicate that MC cancer cells express high levels of several goblet cell markers. AC cancer cells express high levels of enterocyte markers and stem cell markers. This difference may be due to the different origins of cancer cells. To better understand this interesting finding, we performed trajec-tory analysis of the epithelial cells (cancer cells, enterocytes and goblet cells) based on the Monocle 2 algorithm to order the epithelial cells and determine their developmental trajectories (Supplementary Fig. 9a, b, c). This analysis indicated that the developmental status of AC cancer cells and most epithelial cells were the same at the terminal state, whereas most MC cancer cells and goblet cells showed the same developmental status at the beginning of the trajectory path (Fig. 7a. Supplementary Fig. 9d, e). Subsequently, we investigated the transcriptional changes associated with transitional states (Fig. 7b). The expression levels of genes related to protein secretion, regulation of protein kinase activity and regulation of cell adhesion were significantly reduced, and those of genes related to cell division, extracellular matrix organization, myeloid leukocyte-mediated immunity and neutrophil-mediated immunity were significantly increased, along the trajectory. The top thirty and eight DEGs along the trajectory are shown in Fig. 7c, d. *BCAS1*, *HERPUD1*, *CLCA1* and *SLC12A2* were significantly upregulated at the beginning of the trajectory path, whereas *CEACAM7*, *CFTR*, *CLCA4* and *LAMA3* were significantly upregulated at the terminal state.

Our results showed that MC cancer cells have goblet cell-like gene expression characteristics and that their developmental

trajectories parallel those of goblet cells. To obtain a compre-hensive understanding of the correlation between MC cancer cells and goblet cells, we analysed the expression of eight canonical colon goblet cell markers over pseudotime (Fig. 8a). As predicted, the expression of all markers was significantly reduced along the trajectory. The expression of *CLCA1* and *ITLN1* was slightly upregulated at the beginning of the trajectory path. In particular, the expression of *MUC2*, *TFF3*, *SPINK4* and *REG4* was significantly upregulated at the beginning of the trajectory path. Furthermore, hierarchical clustering analysis of cancer cells and goblet cells according to the expression levels of the 8 markers demonstrated similar results (Fig. 8b). A considerable proportion of MC cancer cells clustered together with goblet cells, which specifically expressed high levels of *REG4*, *SPINK4* and *MUC2*.

Then, we analysed the expression of the mucin family genes (*muc1-muc25*) at the single-cell level. The *MUC* family genes were specifically highly expressed in cancer and epithelial cells, and *MUC2* was the most highly expressed *MUC* family gene, especially in MC cancer cells and some epithelial cells (Supplementary Fig. 10a). *MUC* genes were rarely expressed in fibroblasts, B cells, endothelial cells and myeloid cells (Supple-mentary Fig. 10a–e). *MUC2* expression in MC cancer cells was significantly higher than that in AC cancer cells and was similar to that in goblet cells (Fig. 8c, d; Supplementary Data 10). To dissect the regulatory mechanism of *MUC2*, we obtained a dataset of *MUC2* transcription regulating factors from public databases and ranked the factors according to the strength of regulatory effects. Subsequently, we assessed the expression of the top 34 transcription factors along the trajectory (Fig. 8e). TFF3 and FOS expression were significantly reduced along the trajectory, suggesting that they may be involved in the regulation of *MUC2* in MC cancer cells. To further clarify the differences in regulatory mechanisms between AC and MC, we performed a transcription regulation analysis and listed the top 50 results according to regulation strength (Fig. 8f, g). The regulatory network of AC was mainly centred on *PBL6*, *SUCLG1*, *TFF3*, *HMGB1*, *HMGB2* and *PBS4X*, while that of MC was mainly centred on *RPS4X*, *TFF3*, *HMGB2* and *RPL6*. These results demonstrate the similar regulatory effects of *HMGB2* and *MKI67* and *TOP2A* (proliferation-related genes) between the two groups. Particularly in the MC group, *TFF3* regulated several canonical markers of goblet cells, such as *MUC2*, *SPINK4*, *REG4*, *AGR2* and *FCGBP*.

**Cancer-related fibroblast (CAF) features in MC**. Fibroblasts were clustered into eight subclusters by unsupervised clustering. Of these 8 clusters, 2 were related to fibroblasts and 6 were related to myofibroblasts (Fig. 9a). The origins and functions of these subclusters were identified through specific gene expression analysis (Fig. 9b). The fibroblast-1 cluster was characterized by *CCL13*, *CCL11* and *ADH1B* expression, indicating that the cells in this cluster may be involved in the inflammatory response. Fibroblast-2 specifically expressed multiple matrix metallopro-teinases, such as *MMP10*, *MMP3* and *MMP1*. Myofibroblast-1 was characterized by *CST1* and *RGS16* expression. Myofibroblast-2 specifically expressed multiple *WNT* pathway regulatory genes, such as *ASPN*, *GREM1* and *SFRP2*. Myofibroblast-3 was char-acterized by *AREG* and *ITGA8* expression. Myofibroblast-4 showed a high proliferation rate and expressed cell proliferation markers (such as *TOP2A* and *MKI67*). Myofibroblast-5 expressed canonical myofibroblast markers. Myofibroblast-6 was char-acterized by *PDK4*, *GPX3* and *NRG1* expression. The proportions of all eight subclusters in each lesion are provided (Fig. 9c). There were significantly higher proportions of fibroblast-1 and myofibroblast-6 cells in AC than in MC (Fig. 9d; Supplementary

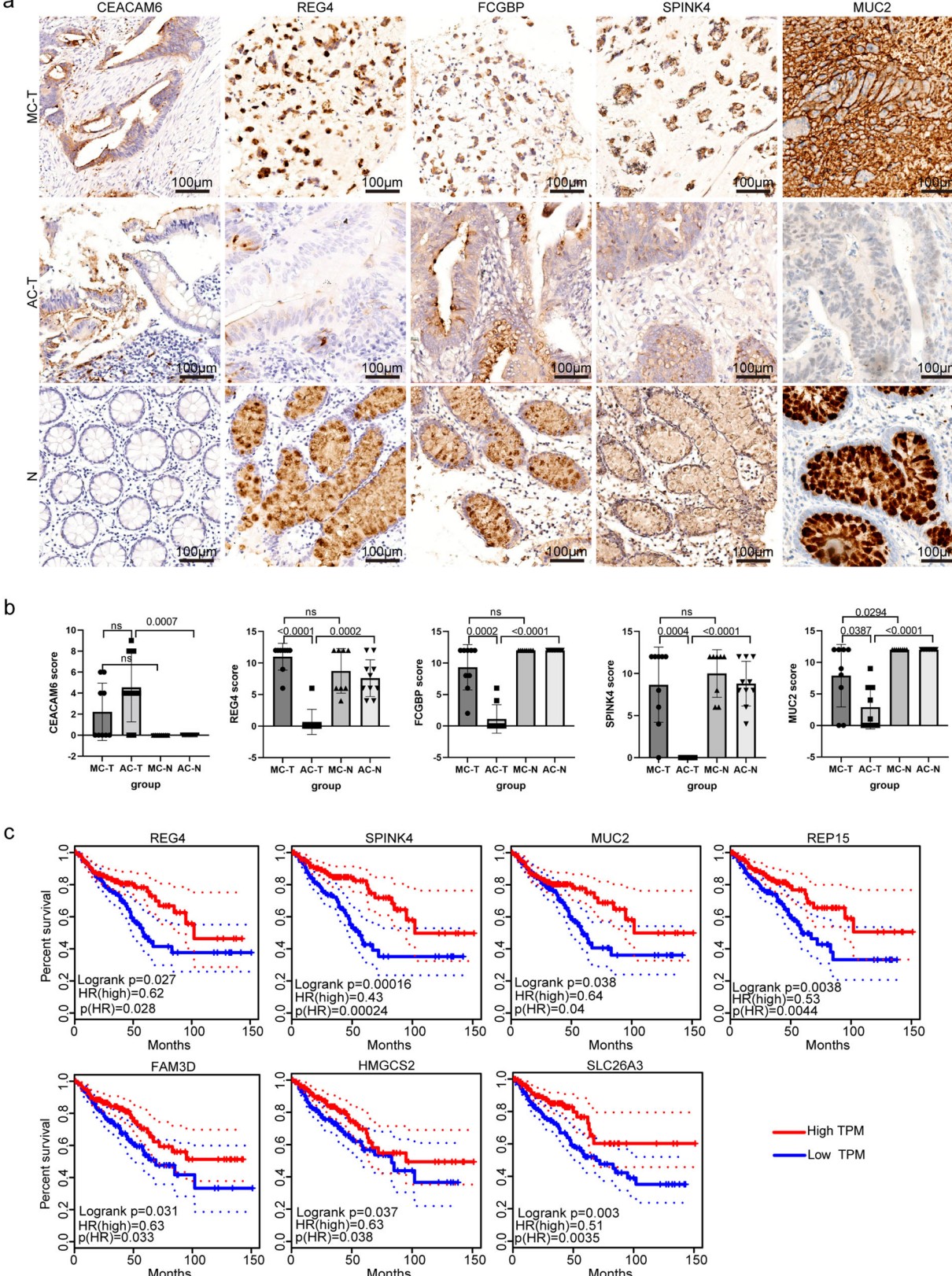

**Fig. 5 IHC and survival analysis of MC markers. a** Representative images of IHC staining of CEACAM6, REG4, FCGBP, SPINK4 and MUC2 expression in paired MC or AC cancer tissues and corresponding normal tissues. Scale bar, 100 μm. **b** Differential analysis of CEACAM6, REG4, FCGBP, SPINK4 and MUC2 levels in paired MC cancer tissues and normal tissues ($n = 9$ biologically independent patients) and AC cancer tissues and normal tissues ($n = 9$ biologically independent patients). Statistical analyses were performed by a nonparametric test followed by the Mann–Whitney test. All the bars represent the mean ± S.D. **c** Kaplan–Meier survival curves were generated for the top 20 DEGs by comparing groups of high (red line) and low (blue line) gene expression. The dotted line represents the 95% confidence interval. $p < 0.05$ according to the log-rank test.

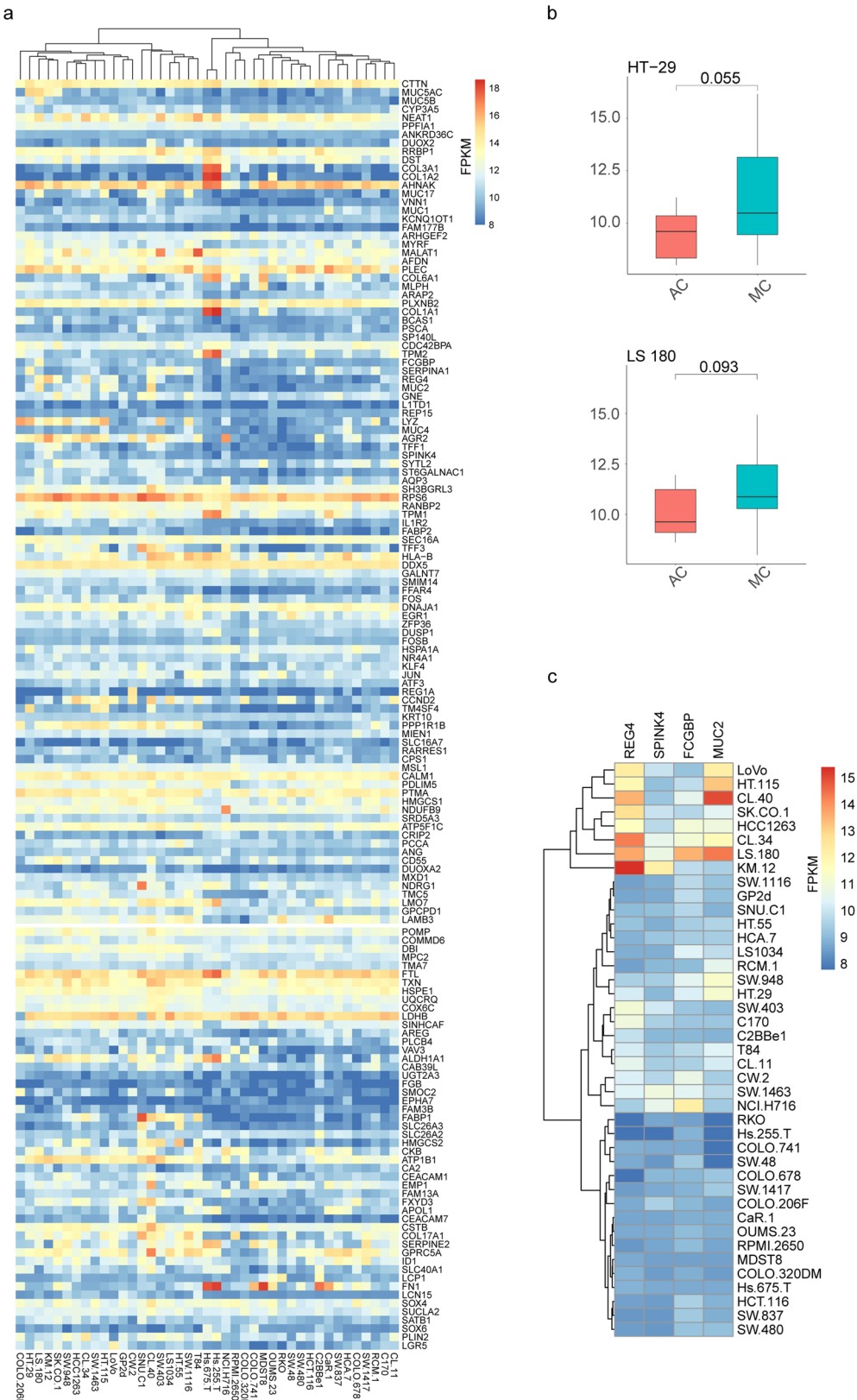

**Fig. 6 Phenotypic predisposition to mucinous adenocarcinoma in 41 cell lines. a** Hierarchical clustering heatmap of the expression of MC and AC module genes in 41 cell lines. **b** Boxplots of the differential expression of MC and AC module genes in HT-29 and LS 180 cell lines. The lower hinge, middle line, and upper hinger of boxplots represented the first, second, and third quartiles of the distributions. The upper and lower whiskers corresponded to the largest and smallest data points within the 1.5 interquartile range. **c** Hierarchical clustering heatmap of the expression of goblet marker genes in 41 cell lines. All colour keys from blue to red represent the FPKM levels from low to high.

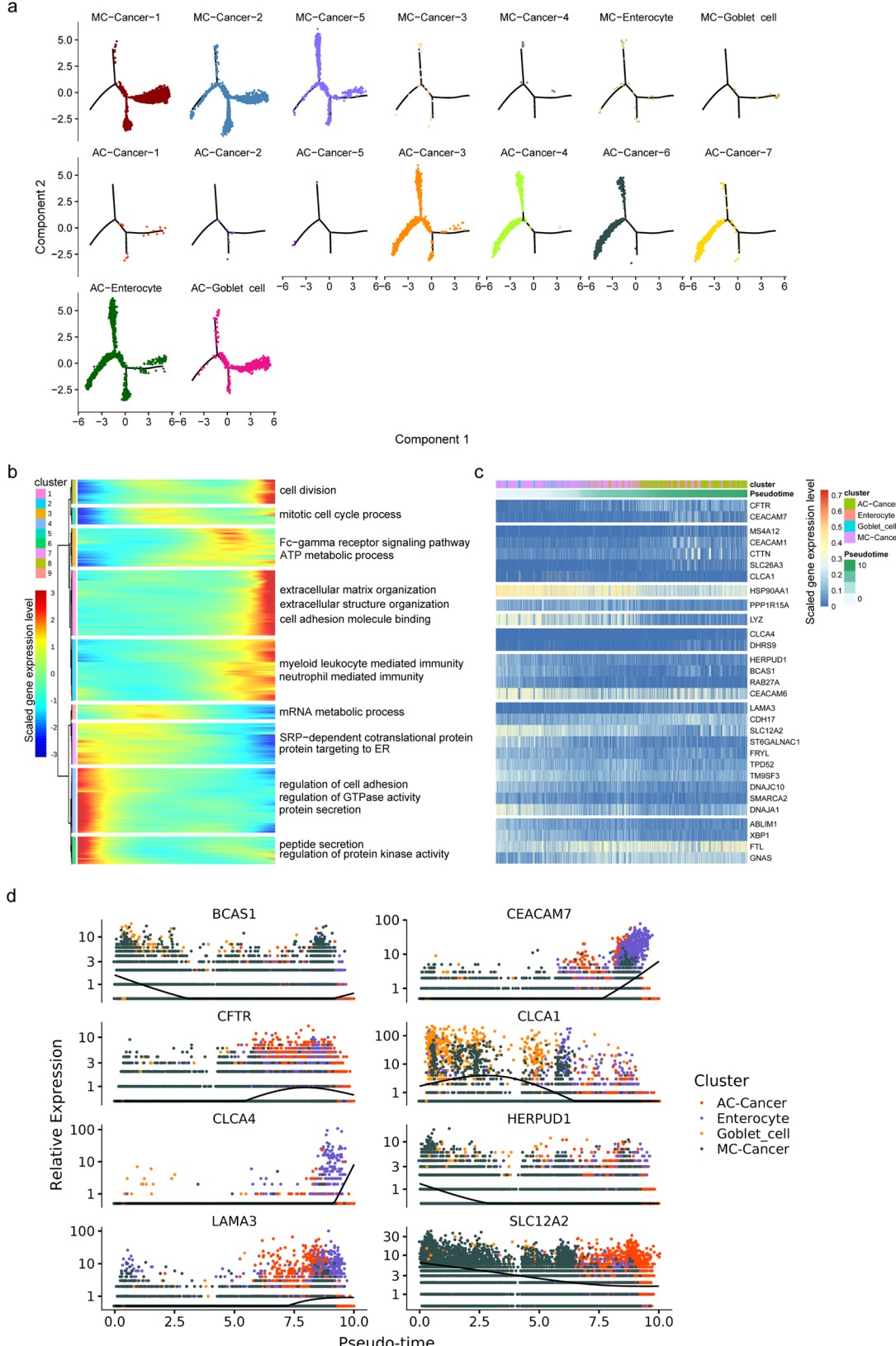

**Fig. 7 Trajectory analysis of epithelial cells in MC and AC samples. a** Distribution diagram of each cell type in the pseudotime trajectory. **b** The DEGs were hierarchically clustered into nine subclusters along the pseudotime trajectory. The top annotated GO terms in each cluster are provided. **c** Heatmap of the gene expression of the top 30 DEGs (in rows) along the pseudotime trajectory of the epithelial cells. The genes were grouped into nine clusters based on their expression patterns. **d** Expression of the top eight DEGs over pseudotime. The different colours indicate different cell types. All colour keys from blue to red represent the scaled gene expression levels from low to high.

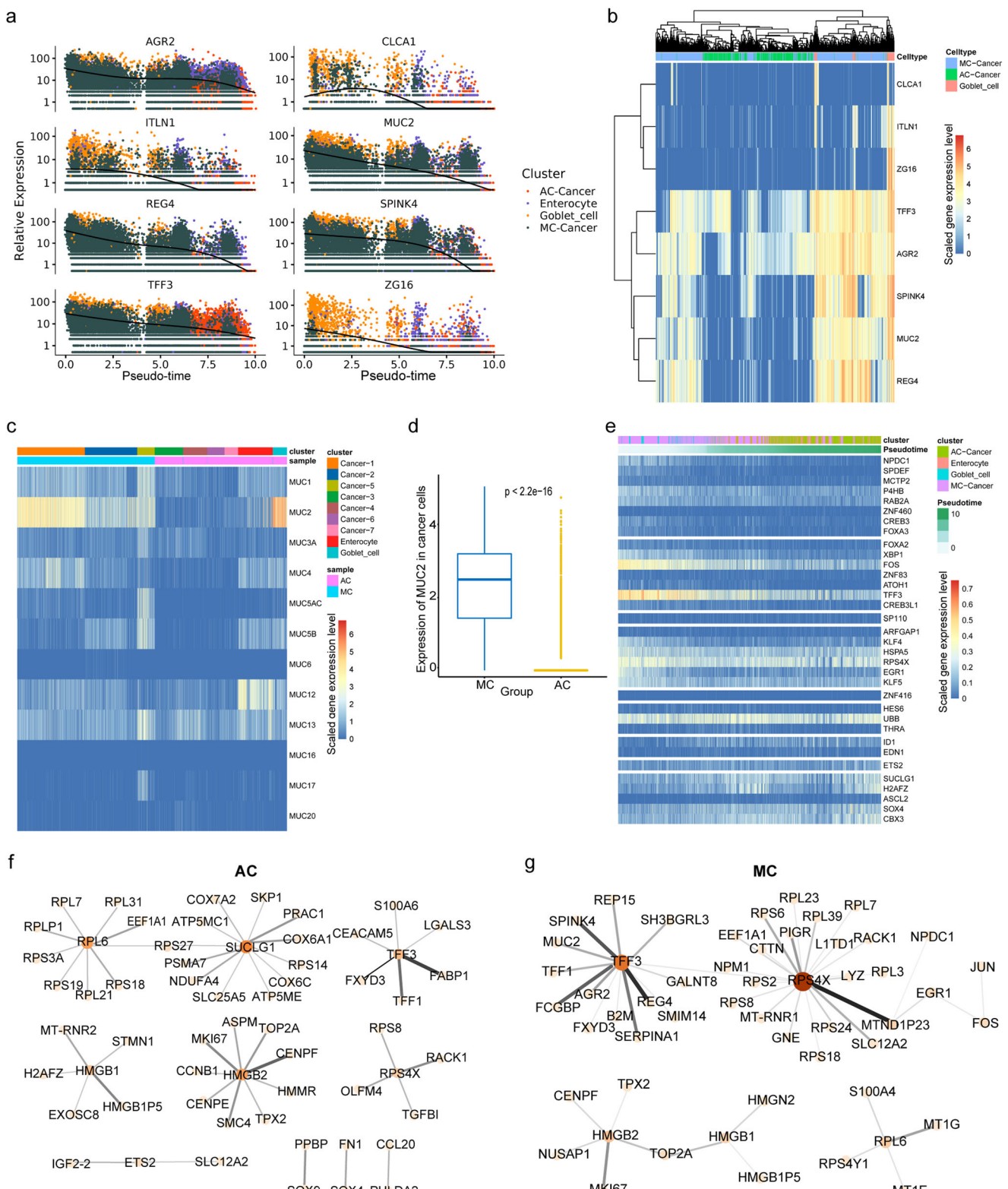

Data 11). Competitive GSVA analysis (Fig. 9e) showed that fibroblast-1 showed enrichment of the *IL6-JAK-STAT3* signalling pathway; fibroblast-2 and myofibroblast-6 showed enrichment of *TNFα* via the *NF-kB* and hypoxia signalling pathways; myofibroblast-1 showed enrichment of the oxidative phosphorylation signalling pathway; myofibroblast-2 showed enrichment of epithelial–mesenchymal transition; myofibroblast-3 showed enrichment of the *WNT/β-catenin*, *TGF-β* and *NOTCH* signalling

pathways; myofibroblast-4 showed enrichment of the *E2F*, *G2M* checkpoint and *MYC* signalling pathways; and myofibroblast-5 showed enrichment of myogenesis signalling.

**Unique intercellular networks of MC.** The crosstalk between different cell types in the TME is a crucial factor leading to tumour progression. We performed cell–cell interaction analyses for AC and MC. Both cancers presented prominent interactions

**Fig. 8 Goblet cell characteristics and regulation of mucin in mucinous adenocarcinoma. a** Expression of eight goblet cell markers over time. Different colours indicate the different cell types. **b** Heatmap displaying the expression levels of 8 colon goblet cell markers in cancer cells and goblet cells. The arrangement of the cells at the top is based on their similarity, defined using hierarchical clustering. **c** Heatmap displaying *MUC* family expression in cancer and epithelial cells in MC and AC. **d** Boxplot indicating the differential expression of *MUC2* in cancer cells between MC and AC. The *p* value was calculated from a *t*-test. The lower hinge, middle line, and upper hinger of boxplots represented the first, second, and third quartiles of the distributions. The upper and lower whiskers corresponded to the largest and smallest data points within the 1.5 interquartile range. **e** Heatmap of the gene expression profiles of the top 34 transcription factors associated with *MUC2* (in rows) along the pseudotime trajectory of epithelial cells as indicated. These genes were grouped into nine clusters based on their expression patterns. Transcriptional regulatory network in AC **(f)** and MC **(g)**. The node colour depth is proportional to the number of neighbours (interacting genes) of each node within each connected network. The line width and colour are proportional to the number of interacting genes. All colour keys from blue to red represent the scaled expression levels of genes from low to high.

between endothelial cells, pericytes and fibroblasts (Fig. 10a, b). We observed enhanced interactions between myeloid cells, endothelial cells, pericytes and fibroblasts, as well as cancer cells and fibroblasts, in MCs compared to ACs.

By comparing the interaction networks of AC and MC, we found that the communication of cancer cells with fibroblasts was significantly increased in MC. To further reveal the regulatory interactions between cancer cells and fibroblasts, CellphoneDB interaction analysis was conducted to explore cell−cell crosstalk in a repository of ligands, receptors, and their interactions (Fig. 10c, d). The *EGFR–COPA* and *NRG1–LGR4* pairs were expressed at higher levels in AC tumours than in MC tumours. We found enhanced *VEGFA–FLT1* signals between cancer cells and fibroblasts, which play an important role in tumour angiogenesis and progression, in MC [35]. Furthermore, MC featured enhanced antiapoptotic activity and signals, as indicated by enhancement of the *TNFSF10–RIPK1*[36] and *TNFRSF11B–TNFSF10* interactions[37]. As expected, although some receptor–ligand pairs were identical in MC, they showed different patterns of interaction between cancer cells and fibroblasts. These results indicate that the crosstalk between cancer cells and fibroblasts via diverse receptor–ligand signals may exert a profound effect on the unique phenotype of MC.

## Discussion

To our knowledge, this is the first study to compare the transcriptomes of AC and MC at the single-cell level. In the present study, we assessed the comprehensive landscape of cancer cells, stromal cells and immune cells in classical and mucinous adenocarcinomas by scRNA-seq analysis. We paid particular attention to MC, the pathogenesis mechanism and biological characteristics of which are largely unknown. Here, we demonstrated that MCs have different biological characteristics from ACs, including different cell subpopulation compositions, cancer cell characteristics and intercellular networks. Notably, our results indicate that MC cancer cells have goblet cell-like properties, as they express high levels of goblet cell markers (*REG4*, *SPINK4*, *FCGBP*, and *MUC2*) and overlap with goblet cells in terms of their developmental trajectories. In addition, upregulation of *TFF3*, *MUC2*, *FCGBP* and *REG4* is the main cause of mucus formation, and *TFF3* plays a vital role in the regulation of mucus.

The characteristics of cancer cells shape the composition of the TME through different mechanisms and lead to different biological behaviours and treatment responses. In the present study, unsupervised clustering of cancer cells was performed according to specific patient traits, and the results were aligned with those of recent research on patient-derived spheroid cultures of CRC[38]. MC cancer cells presented a unique expression pattern, specifically expressing several goblet cell markers (*REG4*, *FCGBP*, *MUC2*, and *CLCA1*) and secreted mucin genes (*MUC2*, *MUC5AC*, and *MUC5B*) (Fig. 3b). GSVA did not identify a consistent function of cancer cell subpopulations in MCs or ACs (Fig. 3d). This is probably due to intratumoral heterogeneity, as indicated by Sophie's study of intratumoral heterogeneity in CRC

at the single-cell level[39]. The inferred CNA profiles were also similar, and no difference in intratumoral heterogeneity was observed between the two groups. These findings suggest that MC cancer cells may have important intrinsic relationships with goblet cells and that their phenotypes may be influenced by factors beyond genomic alterations.

Subsequently, we compared the differences in cancer cells between the two groups at the transcriptional level. The results demonstrated that the top 4 upregulated genes were *REG4*, *SPINK4*, *FCGBP* and *MUC2* (Fig. 4a). Similarly, these were also the genes most associated with the MC phenotype by the unsupervised algorithm. Previous studies confirmed that *MUC2* was the most significantly differentially expressed gene between MC and AC[12,13]. In this study, although MUC2 appeared to be the most differentially expressed protein in MC, much mucus was also stained because MUC2 is a component of mucus molecules (Fig. 5a). In the present study, *REG4* was the most significantly differentially expressed gene and was more frequently expressed in CRCs with mucinous components than in those without mucinous components ($p < 0.001$)[40]. Similarly, Sabine et al.'s research confirmed that *REG4* is more strongly expressed in colorectal tumours (particularly in mucinous carcinomas) than in normal colon tissues and that *REG4* mRNA-positive tumour cells display mucous-secreting, enterocyte-like or undifferentiated phenotypes[41]. In this study, REG4 was the most differentially expressed gene between MC and AC at the protein level ($p < 0.0001$), while MUC2 was the least differentially expressed gene among these four genes at the protein level ($p = 0.0387$) (Fig. 5b). *REG4*[31], *SPINK4*[20] and *MUC2*[20] are canonical colon goblet cell markers. *FCGBP* is a marker of goblet cells in the small intestine[19]. To assess the goblet cell-like molecular features associated with the progression of MC, we conducted trajectory analysis of scRNA-seq data, which allowed us to identify the developmental relationships and the gene expression profiles along the developmental path of the cancer. We found that most MC cancer cells and goblet cells were the same in the early stage of the trajectory path (Fig. 7a; Supplementary Fig. 9d, e). A previous study indicated that the expression of REG4 may be an early event in CRC carcinogenesis according to the results of IHC staining of REG4 in whole tissue sections[42]. In addition, signalling pathways, including those related to protein secretion, peptide secretion, and regulation of protein kinase activity, were upregulated at the beginning of the trajectory path (Fig. 7b). This result is consistent with the mucus characteristics of MC and proves the robustness and reliability of the pseudotime reconstruction model. Importantly, the expression of 8 markers of canonical colon goblet cells was significantly reduced along the trajectory (Fig. 8a). In addition, these markers were highly expressed at the beginning of the trajectory, with substantial overlap between MC cancer cells and goblet cells. Furthermore, we found that a considerable portion of MC cancer cells clustered together with goblet cells and specifically expressed high levels of *REG4*,

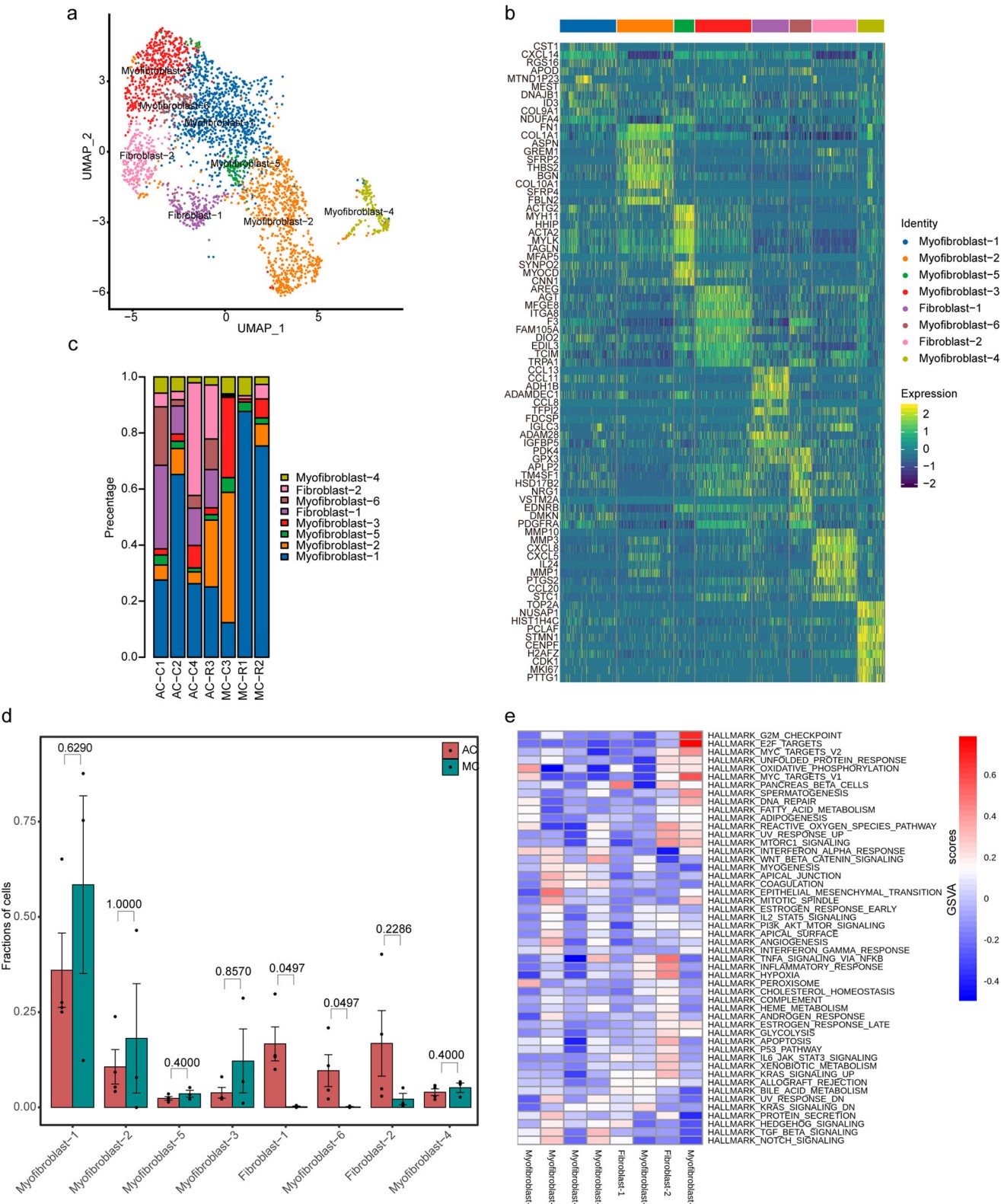

**Fig. 9 Single-cell transcriptomic analysis of cancer-related fibroblasts (CAFs). a** UMAP visualization of CAF clusters. **b** Top 10 genes expressed in 8 CAF cell subclusters. Colour key from blue to yellow represents the scaled expression levels of cell type-specific marker genes from low to high. **c** Distribution of the 8 CAF clusters among 7 samples. **d** Histogram showing the proportions of CAF subclusters in ACs (red) and MCs (green). The analysis was performed using unpaired two-tailed Wilcoxon rank-sum tests, and statistical significance was set at *p* < 0.05. The error bars represent mean ± std. **e** Cluster heatmap of GSVA using 50 hallmark genes from MSigDB among the 8 CAF subclusters. Colour key from blue to red represents the GSVA *scores* from low to high.

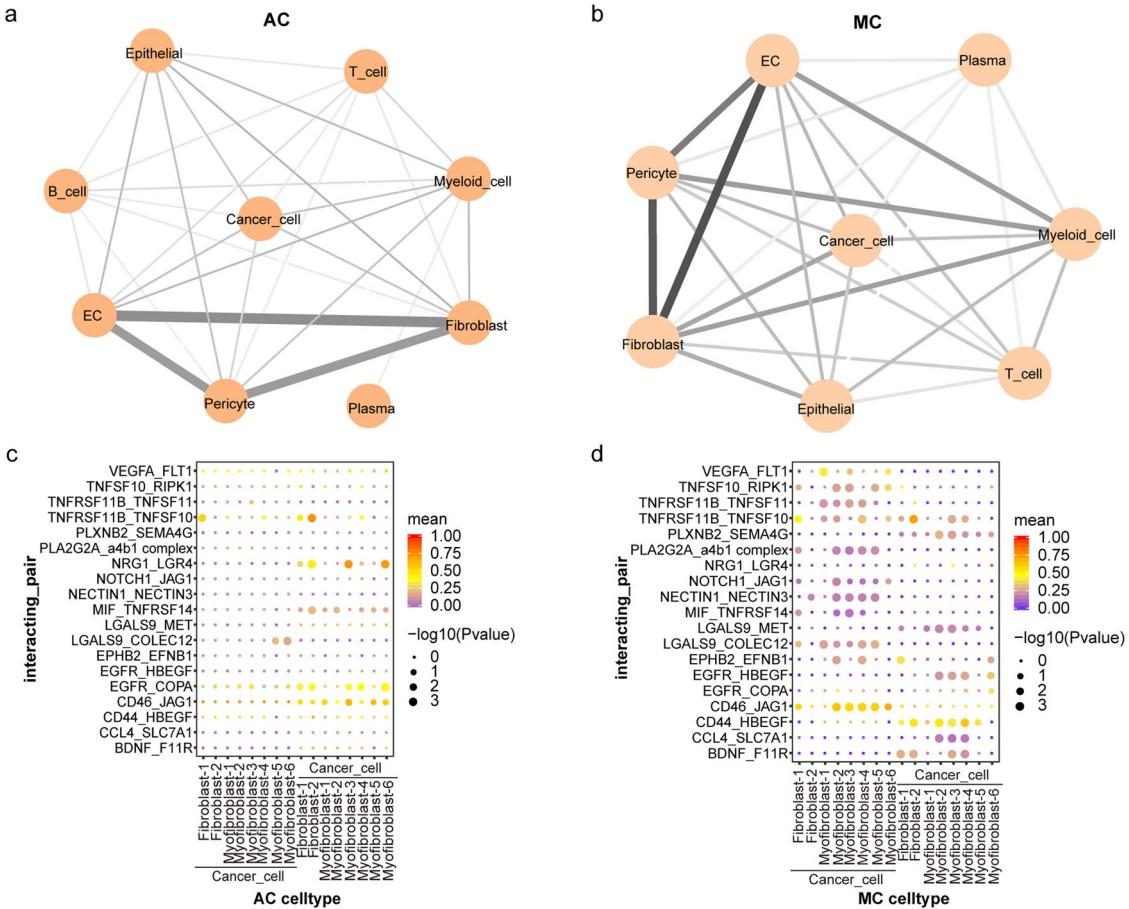

**Fig. 10 Cell–cell communication in MC and AC.** The cellular interaction network among different cell types in AC **(a)** and MC **(b)**. The line width and colour are proportional to the number of interactions between cell types. Selected ligand–receptor interaction pairs in AC **(c)** and MC **(d)**. Colour key from blue to red represents mean expression of two genes in the interacting pair from low to high.

*SPINK4* and *MUC2*, according to the hierarchical clustering analysis (Fig. 8b). Dalerba et al. identified multiple analogous lineages in human colon epithelium and human colorectal benign and malignant tumours, and one of these lineages showed a *MUC2*+, *TFF3*high, *SPDEF*+, and *SPINK4*+ phenotype and a morphology consistent with that of the goblet-like cells[43]. In the present study, we found that MC cancer cells presented molecular characteristics similar to those of goblet cells and specifically expressed high levels of *REG4*, *SPINK4* and *MUC2* compared to AC cancer cells. Dalerba performed the high-throughput parallel analysis based on flow cytometry sorting single cells, screening only approximately 230 genes[43]. In the past 5 years, the maturity of single-cell technology has enabled the robust and reliable detection of a large number of genes and the identification of cell clusters on the basis of different markers.. The goblet cell-like characteristics of MCs are reminiscent of those of goblet cell tumours of the appendix, exhibiting mucinous differentiation[44]. These tumours may originate from pluripotent progenitor stem cells at the base of the intestinal crypt that undergo mucinous differentiation[44,45]. The 2019 World Health Organization classification replaced the name "goblet cell carcinoid" with the more appropriate "goblet cell adenocarcinoma" because the tumour is not a conventional colorectal adenocarcinoma[46]. Taken together, our results confirm that MC cancer cells exhibit goblet cell-like characteristics and may arise from goblet cell progenitor cells or pluripotent stem or progenitor cells, as does goblet cell adenocarcinoma of the appendix. Indeed, the organization and structures of the

colon are similar to those of the appendix. Furthermore, *REG4*, *SPINK4*, *FCGBP* and *MUC2* are promising markers for the diagnosis and treatment of MC.

To date, few studies have explored the mechanism of mucus production in MC. Although the specific molecular characteristics of MC have been well described, the mechanism of mucus production is still unknown. In the present study, signalling pathways, including those related to *SRP*-dependent cotranslation and protein targeting to the *ER*, were upregulated at the beginning of the trajectory path, suggesting that they may play a vital role in mediating mucus synthesis of MCs (Fig. 7b). Furthermore, we focused on *MUC2*, one of the most significantly differentially expressed genes (Fig. 8c, d) that may contribute to mucus production. We found that the *MUC2*-related transcription factors TFF3 and FOS were significantly reduced along the trajectory (Fig. 8e). Furthermore, TFF3 is a hub transcription factor and regulates several canonical markers of goblet cells, including *MUC2*, *SPINK4*, *REG4*, *AGR2* and *FCGBP* (Fig. 8g). TFF3 is a member of the trefoil factor family, and has recently been described as a transcriptional regulator[47]. TFF3 is primarily considered a secretory peptide involved in mucosal protection and defence[27,48], and is a typical product of intestinal goblet cells and most other cells in the mucous epithelium and glands. Previous studies have shown that TFF3 mainly forms a heterodimer connected by disulfide bonds with FCGBP in the intestine[49]. Proteomic analyses identified the binding between MUC2 and FCGBP in the human intestinal mucus layers[50]. In addition, Yu et al. confirmed that the C-terminal domain of MUC2 can form a

heteropolymer with the C-terminal domains of FCGBP and TFF3 in soluble mucus[51]. One of the important member of the transcription factor AP-1 complex is FOS. JUN, FOS, ATF and MAF protein families members together form AP-1, a dimer complex[52]. Limited evidence has indicated that AP-1 mediates the upregulation of *MUC2* at the transcriptional level[53,54]. In addition, compared to AC cancer cells, MC cancer cells exhibited specific enrichment of certain signalling pathways, including the *TNFα* via *NF-kB* and oestrogen response early signalling pathways (Fig. 4b). In a previous study, *TNF-α* was reported to upregulate the transcription of *MUC2* via the *PI3K/AKT/NF-κB* signalling pathway[55]. In HT29-MTX cells (mucus-producing intestinal epithelial cells), oestrogen treatment resulted in a nearly 50% increases in mucin and twofold and eightfold increases in mucus viscosity and elasticity, respectively, compared with no oestrogen exposure[56]. This effect of oestrogen seems to be in line with the higher incidence of MC in female patients in the clinic. Taken together, our results strongly suggest that MC is characterized by mucus mainly due to the upregulation of *TFF3*, *MUC2*, *FCGBP* and *REG4* and that *TFF3* is essential for the transcriptional regulation of these molecules. Furthermore, signalling pathways including those related to *SRP*-dependent cotranslation, protein targeting to the *ER*, *TNF-α* signalling via *NF-κB* and the early oestrogen response may mediate mucus synthesis and secretion.

Transcription regulation analysis showed different regulatory networks in MCs and ACs. Although several hub transcription factors overlapped, the regulatory targets were significantly different (Fig. 8f, g), for example, TFF3 (as described above). In addition, *RPS4X*, another overlapping hub gene, is known to mainly regulate *RACK1*, *TGFBI*, and *OLFM4* in AC. *RACK1* is thought to be an oncogene in colon cancer, and *RACK1*-induced autophagy promotes the survival and proliferation in colon cancer cells[57]. *TGFBI* promotes tumorigenesis by stimulating angiogenesis[58]. *OLFM4* inhibits colon cancer progression as a negative regulator of the *WNT/β-catenin* signalling pathway[59]. Transcriptional upregulation of *RPS4X* is more enriched in MC. We found an enhanced regulatory relationship between *RPS4X* and *MTND1P23*, a pseudogene that is traditionally considered nonfunctional. However, with technological advances, the function of pseudogenes in diseases (especially in cancers) has gradually been revealed[60,61]. More research is needed to illustrate the function of *MTND1P23*. Notably, we found that *RPS4X* had multiple regulatory partners within ribosomes, the organelles that catalyse protein synthesis, including *RPS6*, *RPS2*, *RPS24*, *RPL23*, *RPL39* and *RPL3*. As such, *RPS4X* may cooperate with *TFF3* to catalyse mucus synthesis. Taken together, our results indicate that different transcriptional regulatory networks are active in MC and AC. Importantly, our results strongly suggest that the regulatory networks related to *TFF3* and *RPS4X* are essential for the mucus synthesis.

Fibroblasts provide the stromal structure of the TME and are considered to be involved in paracrine interactions with cancer cells. Fibroblasts are a highly versatile cell type and exhibit extensive heterogeneity[62]. Indeed, 3108 fibroblasts were clustered into 8 clusters, which showed different genetic and functional characteristics (Fig. 9b, e). Importantly, we found an absence of fibroblast-1 and myofibroblast-6 cells in MCs (Fig. 9d). Fibroblast-1 expressed high levels of *CCL13*, *CCL11* and *CCL8*, which are chemoattractants for eosinophils[63–65]. Fibroblast-1 presented an inflammatory phenotype and may play an important role in intestinal immune regulation. CAFs exhibiting an inflammatory phenotype (iCAFs) have been identified within multiple cancers, including ovarian, pancreatic, breast, and skin cancers[66–69]. Myofibroblast-6 expressed high levels of several

metabolism-related genes, including *PDK4*, *GPX3* and *HSD17B2*. Previous evidence supports a key role of CAFs as regulators for metabolic processes in cancer[70]. There is a significant correlation between the intracellular metabolic status of cancer cells and adjacent CAFs in human breast cancer[71]. However, the precise role of the two fibroblast clusters in CRC remains uncertain and requires further in vitro study. Overall, our results suggest that fibroblast-1 (*CCL13+, CCL11+, CCL8+*) and myofibroblast-6 (*PDK4+, GPX3+, HSD17B2+*) cells may be useful markers for distinguishing MCs from ACs.

In conclusion, this study demonstrated many differences between MC and AC by single-cell RNA transcriptomics. Several markers and specific fibroblast clusters were identified. Additionally, our data reveal communication between transcriptional regulatory networks across cancer cells and fibroblasts in MCs and ACs. Importantly, our results highlight the goblet cell-like characteristics of MC cancer cells and describe the mechanism of mucus synthesis and secretion from multiple perspectives. Overall, our study provides unique perspective on understanding MC.

## Methods

**Ethics approval and consent to participate**. The study was approved by the Ethics Committee of Peking University Cancer Hospital & Institute. All patients provided written informed consent for transcriptomic analysis of their lesions as well as participation in the study.

**Sequenced patients**. Seven patients hospitalized from October 2020 to February 2021 at Peking University Cancer Hospital were prospectively enrolled in the study, which was approved by the Peking University Cancer Hospital Ethics Committee. Each patient provided written informed consent. All patients (five males and two females, age range 39–65 years) were diagnosed with nonmucinous adenocarcinoma or mucinous adenocarcinoma according to the consensus standard[2]. All samples for scRNA-seq were obtained from the primary tumour sites; of the patients who provided samples, two had received traditional first-line neoadjuvant chemotherapy, and 5 were naïve patients who had not yet received surgical therapy. One patient had liver metastasis, and one had omentum metastasis. Detailed clinical characteristics of the patients are provided (Supplementary Table 1).

**Tissue dissociation and preparation**. Colorectal cancer tissues were stored in GEXSCOPE Tissue Preservation Solution (Singleron) and shipped to the Singleron laboratory with an ice pack. The specimens were washed 3 times with Hanks balanced salt solution (HBSS, Gibco, Cat. No. 14025-076) and shredded into 1–2 mm pieces. Then, the tissue debris was digested with 2 ml GEXSCOPE Tissue Dissociation Solution (Singleron) at 37 °C for 15 min in a 15 ml centrifuge tube (Falcon, Cat. No. 352095) with sustained agitation. Cells were filtered through 40-micron sterile strainers (Falcon, Cat. No. 352340) and centrifuged (Eppendorf, 5810 R) at $300 \times g$ for 5 min. Then, the supernatant was removed, and the pellets were resuspended in 1 ml PBS (HyClone, Cat. No. SA30256.01). To remove the red blood cells, which are frequently a significant proportion of the cells, 2 mL RBC lysis buffer (Roche, Cat. No. 11 814 389 001) was added to the cell suspension according to the manufacturer's protocol. The cells were then centrifuged at $500 \times g$ for 5 min in a microfuge at 15–25 °C and resuspended in PBS (HyClone, Cat. No. SA30256.01). The cell mixture sample was stained with trypan blue (Bio-Rad, Cat. No. 1450013), and the cell count was determined under a microscope (Nikon, ECLIPSE Ts2); then, the cell suspension was adjusted to a concentration of $1 \times 10^5$ cells per ml. When the cell viability exceeded 80%, subsequent sample processing was performed.

**Single-cell RNA sequencing**. Single-cell suspensions with a concentration of $1 \times 10^5$ cells per ml in PBS were prepared. The single-cell suspensions were then loaded onto microfluidic devices, and scRNA-seq libraries were constructed with the GEXSCOPE Single-Cell RNA Library Kit (Singleron Biotechnologies) according to the Singleron GEXSCOPE protocol. The process included cell lysis, mRNA trapping, labelling of cells (barcodes) and mRNA (UMIs), reverse transcription of mRNA into cDNA, amplification, and fragmentation of cDNA. Individual libraries were diluted to 4 nM and pooled for sequencing. Pools were sequenced on an Illumina HiSeq X with 150 bp paired-end reads.

**Primary analysis of raw read data**. An internal pipeline was used to process the raw reads from scRNA-seq and generate gene expression matrix. Firstly, FastQC v0.11.4 (https://www.bioinformatics.babraham.ac.uk/projects/fastqc/) and fastp[72] processed raw reads to remove low-quality reads. Cutadapt[73] was used to trim poly-A tails and adapter sequences. Then, we extracted cell barcodes and UMI

counts. Next, we mapped the reads to the reference genome GRCh38 (Ensembl version 92 annotation) using STAR[74] v2.5.3a. FeatureCounts[75] v1.6.2 software was used to obtain UMI counts and gene counts of each cell, which were used to generate expression matrix files for subsequent analysis.

**Quality control, dimension reduction and clustering**. Prior to analysis, cells were filtered according to the following criteria: UMI count less than 30,000; Gene counts between 200 and 5000; and mitochondrial content more than 50%. After filtering, dimensionality reduction and clustering were performed by using appropriate functions in Seurat v2.3[76]. Then, NormalizeData and ScaleData functions were used to normalized and scaled all gene expression data. FindVariableFeautres function was used to select top 2000 most variable genes for PCA. FindClusters was used to separate the cells into different clusters based on the top 20 principal components. Harmony[77] was used to remove the batch effect. Finally, the cells were ploted in two-dimensional space by applying the UMAP algorithm.

**CNV inference from scRNA-seq data**. We identified malignant cells by inferring large-scale chromosomal copy number variations (CNVs) in each single cell based on a moving averaged expression profile across chromosomal intervals[78–80]. To run inferCNV, we used a hidden Markov model (HMM) to predict the CNV level and implemented the i6 HMM model in inferCNV. To identify CNV differences among cluster, GRCh38 gene information was used to convert each CNV to p- or q-arm format based on its location. After data conversion, we merged the CNVs that belonged to the same arm level. Finally, arm-level CNVs were annotated as a gains or losses. The results were visualized with pheatmap (R package).

**Differentially expressed gene (DEG) analysis**. The Seurat FindMarkers function was performed to identify differentially expressed genes (DEGs), which is based on the Wilcoxon likelihood-ratio test with default parameters. The genes expressed in more than 10% of the cells in a cluster with an average log (fold change) value greater than 0.25 were selected as DEGs. We combined canonical expression markers found in the DEGs with those known from the literature to annotate the cell type of each cluster. The heatmaps/dot plots/violin plots generated by Seurat DoHeatmap/DotPlot/Vlnplot functions visualized the expression of markers of each cell type. Cells expressing markers for multiple cell types were identified as doublet cells and were removed manually.

**Functional gene module analysis**. Hotspot was used to identify functional gene modules that illustrate heterogeneity within cancer subpopulations[81]. Briefly, we used the 'danb' model and selected the top 500 genes with the highest autocorrelation $zscores$ for module identification. Modules were then identified using the create_modules function, with $min\_gene\_threshold = 15$ and $fdr\_threshold = 0.05$. Module scores were calculated by using the calculate_module_scores function.

**Jaccard similarity analysis**. The Jaccard similarity coefficient was calculated to compare the transcriptional similarity between cell types using their signature genes[82]. We evaluated transcriptional similarity between meta-programmes of malignant cells and signatures of cell types/states by calculating *Jaccard similarity coefficients* using the top 50 marker genes.

**Pathway enrichment analysis**. We performed Gene Ontology (GO) and Kyoto Encyclopedia of Genes and Genomes (KEGG) analyses to investigate the potential functions of DEGs with the "clusterProfiler" R package[83]. Pathways with a $p\_adj$ value less than 0.05 were considered significantly enriched. The molecular function (MF), biological process (BP) and cellular component (CC) were used to explore the functions of each object. The 50 hallmark gene sets in the MSigDB database (https://www.gsea-msigdb.org/gsea/msigdb) were used for GSVA pathway enrichment analysis, and the average gene expression of each cell type was used as input data.

**Gene regulatory network inference**. PySCENIC (version 0.11.0) was used to perform single-cell regulatory network analysis. We performed the analysis by following the protocol steps described in the SCENIC workflow[84]. The 'pyscenic grn' function was first used to generate coexpression gene regulatory networks using the 'grnboost2' method. AUCell analysis was further performed using the 'pyscenic aucell' function with the parameters 'rank_threshold' 5000, 'auc_threshold' 0.05 and 'nes_threshold' 3.

**Trajectory analysis**. Cell differentiation trajectories were constructed with Monocle2[85]. DEGs were used to sort cells in order of spatial-temporal differentiation. We used DDRTree and the FindVariableFeatures function to perform dimension reduction. Finally, the trajectory was visualized by the plot_cell_trajectory function.

**Cell−cell interaction analysis**. CellPhoneDB[86] was used to perform cell−cell interaction analysis based on receptor−ligand interactions between two cell types/ subtypes. To calculate the null distribution of average ligand–receptor expression levels of the interacting clusters, we randomly arranged the cluster labels of all cells 1000 times. The threshold for individual ligand or receptor expression was based on the cut-off value of the average log gene expression distribution for all genes across all cell types. A $p$ value < 0.05 and average log expression >0.1 indicated significant cell−cell interactions, and these interactions were visualized with the circlize (0.4.10) R package.

**TCGA survival analysis**. We used the Gene Expression Profiling Interactive Analysis (GEPIA)[87] web server for TCGA survival analysis. Specifically, genes and cancer subtypes of interest were chosen as the inputs to generate the survival curves for patient overall survival (OS) and the statistical testing results. We used the median as the cut-off value to assign patients into the low and high groups. P values < 0.05 were considered statistically significant.

**Histology procedures**. Nine pairs of paired MC tissues (cancer and normal) and nine pairs of paired AC tissues (cancer and normal) were collected (including sequenced patients). Tissues were fixed in 4% formaldehyde solution overnight and embedded in paraffin. Sections were subjected to haematoxylin and eosin (H&E) and IHC staining. All IHC was performed by the Roche platform (BenchMark ULTRA). Tissues were warmed to 72 °C from medium temperature, and then warmed to 100 °C, and incubated for 8 minutes. Then, ULTRA CC1 was applied for 20 min, and ULTRA CC1 was applied for 36 min. Subsequently, titration was performed by applying the primary antibody for 36 min. Counterstaining was performed with haematoxylin II for 4 min. Post counterstaining was performed with bluing reagent for 4 min. The antibodies used were as follows: CEACACAM6 (85102 S, 1:500 dilution, CST, USA), REG4 (ab255820, 1:1000 dilution, Abcam, UK), FCGBP (ab121202, 1:200 dilution, Abcam, UK), SPINK4 (ab121257, 1:100 dilution, Abcam, UK) and MUC2 (ab134119, 1:300 dilution, Abcam, UK). *IHC scores* were independently determined by two experienced pathologists blinded to the clinical and pathological data. The scores were evaluated based on staining intensity and the percentage of positive cells in each of the sections. The staining intensity was scored as follows: 0, no staining; 1, light yellow staining; 2, yellow−brown staining; and 3, deep brown staining. The percentage of positive cells was scored as follows: 1, <10%; 2, 10−49%; 3, 50−74% and 4, 75−100%. The final score was calculated as follows: positive cell score × staining intensity score.

**Statistics and reproducibility**. Statistical analysis for the sequencing data and their criteria for significance are described above. Statistical analysis of other data was performed using GraphPad Prism (version 8). Cell distribution comparisons between two groups were performed using unpaired two-tailed Wilcoxon rank-sum tests. DEGs between two groups and cell types were identified using a Wilcoxon rank sum test. Specific statistical analysis for comparison were described in the corresponding figure legends. Significance of differences was determined as indicated, and differences with $P < 0.05$ were considered statistically significant. AC samples ($n = 4$ biologically independent samples) and MC samples ($n = 3$ biologically independent samples) were obtained for scRNA-seq analyses. MC paired tissues (cancer and paired normal tissues) ($n = 9$ biologically independent patients) and AC paired tissues (cancer and paired normal tissues) ($n = 9$ biologically independent patients) were obtained for IHC analyses.

**Reporting summary**. Further information on research design is available in the Nature Portfolio Reporting Summary linked to this article.

## Data availability

The original transcriptomic data generated during this study are publicly available in National Genomics Data Center (accession ID: HRA003634). Supplementary Data 4 contains the source data for Fig. 2a, b. Supplementary Data 5 contains the source data for Fig. 3f. Supplementary Data 8 contains the source data for Fig. 5b. The source data that support the findings of Fig. 5c are available from GEPIA, http://gepia.cancer-pku.cn/detail.php?gene. Supplementary Data 9 contains the source data for Fig. 6b. Supplementary Data 10 contains the source data for Fig. 8d. Supplementary Data 11 contains the source data for Fig. 9c, d. All other data are available from the corresponding author (or other sources, as applicable) on reasonable request.

## Code availability

No unique code was generated in this study. All software tools used in this study are freely available. The authors declare that all R scripts supporting the findings of this study are available from the corresponding author upon reasonable request.

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

## Acknowledgements

We thank all the patients for their generous donation of tissue samples for analysis in this study. We thank Dianxiang Geng for performing scRNA-seq experiments and discussing the experimental results. This work was supported by the National Natural Science Foundation of China (82173156) and Beijing Hospitals Authority Clinical Medicine Development of Special Funding (ZYLX202116). The funders had no role in the study design, data collection and analysis, decision to publish, or preparation of the manuscript.

## Author contributions

F.J.H., Y.J.L., and L.W. conceived and supervised the study. F.J. drafted the manuscript. Y.J. supervised the sample collection and clinical annotation, with help from X.Z.L. and Y.J.C.. D.B.J. performed the data analysis. L.Z. performed the histology procedures. A.W.W. and L.W. reviewed and edited the manuscript. All authors approved the final manuscript.

## Competing interests

The authors declare no competing interests.
