## [Peer Review File · Communications Biology]

Reviewers' comments:

Reviewer #1 (Remarks to the Author):

The authors analyzed human colorectal mucinous adenocarcinoma using scRNA-seq. The information and knowledge obtained by the authors are important and should be published. I have two suggestions that would make this report even more valuable.

1) The authors should show in the main Figure the representative H&E images of classical adenocarcinoma (AC) and mucinous adenocarcinoma (MC) that were used in this study. The authors also should show in the Supplementary Figure the H&E images of the rest of the samples which were not chosen for the main Figure. This should be easy since the total number of samples is 7. Related to this image presentation, since the authors successfully determined that REG4, SPINK4, FCGBP and MUC2 could be used as promising markers for the diagnosis of MC but not for AC, they should perform immunohistochemistry (IHC) to detect at least one of the four genes in MC but not in AC to further prove their finding at a protein level using both multiple MC and AC samples. Human Protein Atlas posts the information of antibodies working for IHC for all 4 genes, the information of which is available in their website. Testing fibroblast markers, CCL13, CCL11, CCL8, PDK4, GPX3 and/or HSD17B2, that the authors identified in MC but not in AC might also be good for this IHC experiment. The histological image information always helps to get better ideas about diseases beyond the gene expression sequencing data.

2) RNA-seq datasets of colorectal cancer cell lines are publicly available. For example, Zemin Zhang and colleagues analyzed 41 colorectal cancer cell lines by RNA-seq (Klijn et al., Nat Biotechnol 2015). Would it be possible to categorize or characterize these cell lines as to whether these cell lines belong to AC or MC by using the data that the authors obtained in their current scRNA-seq study? This information will help colorectal cancer researchers since cell lines are still useful for drug development and the categorization/characterization helps to know which types of colon cancer the researchers are targeting when they use such colorectal cancer cell lines.

Reviewer #2 (Remarks to the Author):

The manuscript by Hu and colleagues explores single-cell RNA sequencing for understanding cell lineages of mucinous adenocarcinoma (MC) regarding tumor cell features, Cancer-related fibroblast (CAF) features as well as cell-cell communications.

Overall, to my knowledge, single-cell dataset of MC has not already been reported until now. However, the sample size of single-cell cohort is small and no validation experiments (at least using immune staining) for specific findings. In addition, comparative analysis of MC and AC is the key point in this single-cell study but need more detailed analysis. Therefore, I am not entirely convinced that this manuscript should be published in Communications Biology.

Major:

1. The conclusion in the Abstract part is unclear, please indicate the main findings of this study. For example, what is difference in cell composition between MC and AC, what's the phenotype of tumor cells of MC?
2. How this study classified normal epithelial cells from the cancer cells? (page 4, line 96)
3. As sample size of single-cell cohort is limited, authors should carry out validation analysis using an independent cohort with larger samples size. For example, the authors should performed IHC staining using antibody against specific markers for MC and AC tumor cells, according to DEG analysis.
4. Also, as MC and AC are two histologic subtype of CRC, the authors should performed coupled H&E staining. This can improve the quality of this study.
5. To compare the transcriptional difference between MC tumor cells and AC tumor cells, the authors can apply non-negative matrix factorization (NMF) analysis to define underlying transcriptional programs specific to each tumor and identify gene meta-programs that are shared or exclusive for MC and AC.
6. The authors used lineage markers (for sample, MUC2 and GNE for goblet cell) to define the phenotypes of AC and MC cancer cells. This is not sufficient to make a conclusion like that "our

study reveals the goblet cell-like properties of MC cancer cells". The authors should apply kNN or other unsupervised algorithms to define the phenotypes of AC and MC cancer cells.

7. The authors performed survival analysis to show that CRC patients expressing high levels of MC markers such as REG4, SPINK4, MUC2, REP15, FAM3D, HMGCS2 and SLC26A3 experienced prolonged OS (Fig. 3c). However, the authors described that mucinous adenocarcinoma (MC) has a worse overall prognosis in the introduction part (page 2, line 27-28). Please provide a reasonable interpretation.

8. Figure S2-Figure S4 showed the results of B cells, myeloid and endothelium, corresponding description in the manuscript was that "more details are shown in the supplementary materials (Fig.S2-S4)". If the authors want to show these cells, please respect them with more words. I guess that their scRNA-seq sample size is not enough to make a systematic analysis for these cells, so please focus on cancer cells and provide validating assays.

9. Cell-cell communication analysis (Figure 7) is based on ligand-receptor genes, but the authors also used SCENIC analysis to identify the underlying TF. I can't understand why the authors provided such information "(Fig. 7c, d)". What information TF analysis can provide for explaining the difference of cellular communication in AC and MC?

Minor:

1. Page 6, please add refs for these genes (PPBP, CXCL5, and GGH) supporting their function in activation of neutrophils
2. Page 15, line 528, "Harmony" but not "Harnomy"

Reviewer #1:

1) The authors should show in the main Figure the representative H&E images of
classical adenocarcinoma (AC) and mucinous adenocarcinoma (MC) that were used in
this study. The authors also should show in the Supplementary Figure the H&E images
of the rest of the samples which were not chosen for the main Figure. This should be
easy since the total number of samples is 7. Related to this image presentation, since
the authors successfully determined that REG4, SPINK4, FCGBP and MUC2 could be
used as promising markers for the diagnosis of MC but not for AC, they should perform
immunohistochemistry (IHC) to detect at least one of the four genes in MC but not in
AC to further prove their finding at a protein level using both multiple MC and AC
samples. Human Protein Atlas posts the information of antibodies working for IHC for
all 4 genes, the information of which is available in their website. Testing fibroblast
markers, CCL13, CCL11, CCL8, PDK4, GPX3 and/or HSD17B2, that the authors
identified in MC but not in AC might also be good for this IHC experiment. The
histological image information always helps to get better ideas about diseases beyond
the gene expression sequencing data.

**Response:** Thank you for your comments. We have added HE images of 3 MC and 4
AC tissues in this study, images were shown in Figure 1b (page 4, line 98-99). We
selected nine paired tissues (cancer and normal) from MC patients and 9 paired tissues
(cancer and normal) from AC patients, subsequently immunohistochemistry (IHC) was
performed to detect CEACAM6, REG4, SPINK4, FCGBP and MUC2, respectively
(Figure 3e, f) (page 7, line 215-222). As for the fibroblast marker, we did not consider
it to be an ideal marker due to its significantly lower expression in MC cancer cells
compared to AC cancer cells, so immunohistochemical validation was not performed.

**Figure 1b.** Hematoxylin-eosin (H&E) staining for sequenced samples. Scale bar, 200 μm.

**Figure 3e.** Representative images of CEACAM6, REG4, FCGBP, SPINK4 and MUC2 expression
 from immunohistochemistry staining in paired MC, AC cancer tissues and their normal tissues,
 Scale bar, 100µm.

**Figure 3f.** Differential analysis of CEACAM6, REG4, FCGBP, SPINK4 and MUC2 level in paired
 MC cancer tissue and their normal tissues(n=9), AC cancer tissues and their normal tissues(n=9).
 Statistical analyses were performed by non-parametric test followed by Mann-Whitney test. All the
 bars represent mean ± S.D.

2) RNA-seq datasets of colorectal cancer cell lines are publicly available. For example,
 Zemin Zhang and colleagues analyzed 41 colorectal cancer cell lines by RNA-seq
 (Klijn et al., Nat Biotechnol 2015). Would it be possible to categorize or characterize
 these cell lines as to whether these cell lines belong to AC or MC by using the data that
 the authors obtained in their current scRNA-seq study? This information will help
 colorectal cancer researchers since cell lines are still useful for drug development and
 the categorization/characterization helps to know which types of colon cancer the
 researchers are targeting when they use such colorectal cancer cell lines.

**Response:** Thank you for your comments. First of all, Hostpot analysis was performed
 to identified these modular genes that belonged to MC or AC according to our data
 (Figure 3c, d) (page 7, line 208-215). Then, we downloaded the RNA-seq data of 41
 colorectal cancer cell lines in the study by Zhang et al. RNA-seq data of module genes
 belonging to MC or AC were screened (Figure 4a). Finally, the expression levels of

genes in the MC or AC groups were analyzed to compare the phenotypic tendencies of
41 cell lines (Supplementary Figure 8, Figure 4b) (page 7, line 227-238).

**Figure 3c, d.** **c** Heatmap of gene correlations, with shades of color representing the Z-score values.
The notes on the left represent gene sets. **d** Heatmap of similarity between cell types and gene sets,
with shades of color representing the Jaccard similarity coefficient.

**Figure 4a.** Hierarchical clustering heatmap of expression of MC and AC module genes in 41 cell

lines.

**Supplementary Figure 8. Differential expression of MC group or AC group genes in 41**
**colorectal cancer cell lines.**

**Figure 4b.** Boxplots of differential expression of MC and AC module genes in HT-29 and LS 180
 cell line.

Reviewer #2:

1) The conclusion in the Abstract part is unclear, please indicate the main findings of
 this study. For example, what is difference in cell composition between MC and AC,
 what's the phenotype of tumor cells of MC?

**Response:** Thank you for your comments. We have indicated the main findings of this
 study in the abstract (page 3, line 41-46). The revised text reads as follows: “We found
 an absence of fibroblast-1 (CCL13+, CCL11+, CCL8+) and myofibroblast-6 (PDK4+,
 GPX3+, HSD17B2+) in MC. Furthermore, our results indicate that MC cancer cells
 have goblet cell-like properties, compared to AC cancer cells they highly express goblet
 cell markers (REG4, SPINK4, FCGBP and MUC2). TFF3 is essential for the
 transcriptional regulation of these molecules, which may cooperate with RPS4X and
 eventually lead to the MC mucus phenotype.”

2) How this study classified normal epithelial cells from the cancer cells? (page 4, line
 96)

**Response:** Thank you for your comments. We have added violin plots of marker gene
 expression, additionally CNV score were also performed to classify normal epithelial
 cells from cancer cells (page 5, line 109-113). The revised text reads as follows: “We
 calculated marker genes to classify normal epithelial cells from cancer cells
 (Supplementary Fig. 2a), additionally CNV score also confirmed the accuracy of the
 classification (Supplementary Fig. 2b), in which cancer cells showed higher CNV score
 than normal epithelial cells.”

**Supplementary Fig. 2. Classified cancer cells from the normal epithelial cells. a** Violin plots
 showing the expression distribution of selected marker genes across epithelial cell clusters. **b** Violin
 plots showing the CNVscore of epithelial cell clusters and myeloid cell clusters.

3) As sample size of single-cell cohort is limited, authors should carry out validation
 analysis using an independent cohort with larger samples size. For example, the authors
 should performed IHC staining using antibody against specific markers for MC and AC
 tumor cells, according to DEG analysis.

**Response:** Thank you for your comments. We selected nine paired tissues (cancer and
 normal) from MC patients and 9 paired tissues (cancer and normal) from AC patients,
 subsequently immunohistochemistry (IHC) was performed to detect CEACAM6,
 REG4, SPINK4, FCGBP and MUC2, respectively (Fig. 3e, f) (page 7, line 215-222).

**Fig 3e.** Representative images of CEACAM6, REG4, FCGBP, SPINK4 and MUC2 expression from from immunohistochemistry staining in paired MC, AC cancer tissues and their normal tissues, Scale bar,
100µm.

**Fig 3f.** Differential analysis of CEACAM6, REG4, FCGBP, SPINK4 and MUC2 level in paired MC
cancer tissue and their normal tissues(n=9), AC cancer tissues and their normal tissues(n=9).
Statistical analyses were performed by non-parametric test followed by Mann-Whitney test. All the
bars represent mean ± S.D.

4) Also, as MC and AC are two histologic subtype of CRC, the authors should
performed coupled H&E staining. This can improve the quality of this study.

**Response:** Thank you for your comments. We have added HE images of 3 MC and 4
AC tissues in this study, images were shown in Fig. 1b (page 4, line 98-99).

**Fig. 1b.** Hematoxylin-eosin (H&E) staining for sequenced samples. Scale bar, 200 µm.

5) To compare the transcriptional difference between MC tumor cells and AC tumor
cells, the authors can apply non-negative matrix factorization (NMF) analysis to define
underlying transcriptional programs specific to each tumor and identify gene meta-
programs that are shared or exclusive for MC and AC.

**Response:** Thank you for your comments. We have performed Hostpot analysis and
Jaccard.Similarity analysis to define underlying transcriptional programs specific to
each tumor and identify gene meta-programs that are shared or exclusive for MC and
AC (Fig. 3c, d) (page 7, line 208-215).

**Fig. 3c, d.** **c** Heatmap of gene correlations, with shades of color representing the Z-score values.

The notes on the left represent gene sets. **d** Heatmap of similarity between cell types and gene sets,

with shades of color representing the Jacquard similarity coefficient.

6) The authors used lineage markers (for sample, MUC2 and GNE for goblet cell) to
 define the phenotypes of AC and MC cancer cells. This is not sufficient to make a
 conclusion like that “our study reveals the goblet cell-like properties of MC cancer
 cells”. The authors should apply kNN or other unsupervised algorithms to define the
 phenotypes of AC and MC cancer cells.

**Response:** Thank you for your comments. We have performed Hostpot analysis and
 Jaccard.Similarity analysis to further define the phenotypes of AC and MC cancer cells
 (Fig. 3c, d) (page 7, line 208-215).

**Fig. 3c, d.** **c** Heatmap of gene correlations, with shades of color representing the Z-score values.

The notes on the left represent gene sets. **d** Heatmap of similarity between cell types and gene sets,

with shades of color representing the Jacquard similarity coefficient.

7) The authors performed survival analysis to show that CRC patients expressing high
levels of MC markers such as REG4, SPINK4, MUC2, REP15, FAM3D, HMGCS2 and
SLC26A3 experienced prolonged OS (Fig. 3c). However, the authors described that
mucinous adenocarcinoma (MC) has a worse overall prognosis in the introduction part
(page 2, line 27-28). Please provide a reasonable interpretation.

**Response:** Thank you for your comments. The prognostic difference between MC and
AC was controversial. We have also explained in the text, corresponding description in
the manuscript was that “Whether the prognosis of MC is different from that of AC is
debated. Some studies have reported that MC is associated with worse survival than
AC^{8 9}, while one study showed MC and AC have similar survival¹⁰; furthermore,
another study showed better prognosis in MC than AC¹¹(page 3, line 57-60).” We are
very sorry for the imprecise concept in the introduction, which has been corrected,
corresponding description in the manuscript was that “The prognostic difference
between MC and AC was controversial (page 3, line 32-35).” In addition, our survival
analysis results are based on the strength of gene expression, while the survival
differences mentioned in this article are based on survival analysis of clinical data.
There is no comparison between survival analyses using different parameters.

8) Figure S2-Figure S4 showed the results of B cells, myeloid and endothelium,
corresponding description in the manuscript was that “more details are shown in the
supplementary materials (Fig.S2-S4)”. If the authors want to show these cells, please
respect them with more words. I guess that their scRNA-seq sample size is not enough
to make a systematic analysis for these cells, so please focus on cancer cells and provide
validating assays.

**Response:** Thank you for your comments. There are several reasons why we did not
delve into B cells, myeloid cells and endothelial cells. First, we found no meaningful
differences between MC and AC in B cells, myeloid cells or endothelial cells at the
time of data mining. Second, we found something worth investigating between MC and
AC cancer cells, so we focused on cancer cells and performed validation experiments.
In addition, the size of the paper is also a factor we consider. There are few single-cell
studies of MC, so we put the preliminary analysis results of b cells, myeloid cells and
endothelial cells into the supplementary materials to allow everyone to more
comprehensively understand the difference between MC and AC.

9) Cell-cell communication analysis (Figure 7) is based on ligand-receptor genes, but
the authors also used SCENIC analysis to identify the underlying TF. I can't understand
why the authors provided such information “(Fig. 7c, d)”. What information TF analysis
can provide for explaining the difference of cellular communication in AC and MC?

**Response:** Thank you for your comments. We have put the content of TF analysis into
Fig. 6f, g (page 9, line 284-292).

**Minor:**

1. Page 6, please add refs for these genes (PPBP, CXCL5, and GGH) supporting their
function in activation of neutrophils

**Response:** Thank you for your comments. Refs for genes (PPBP and CXCL5)
supporting their function in activation of neutrophils have been added (page 7, line 200-
201).

2. Page 15, line 528, “Harmony” but not “Harnomy”

**Response:** Thank you for your comments. Misspellings have been corrected.

Reviewers' comments:

Reviewer #1 (Remarks to the Author):

The authors addressed my suggestions sufficiently, including H&E images, IHC and cell line analysis. I have two minor comments.

Comment 1. The authors list TFF3 as a transcription factor, but to my knowledge and my quick Google and PubMed searches, TFF3 is considered to be a secreted protein but not an established transcription factor. There is a possibility that I might have missed important papers on TFF3 as a transcription factor though. But if my knowledge and searches are correct, the authors need to rewrite a part of the text and reanalyze their data, especially Figure 6. And I recommend all of the transcription factors that the authors listed in Figure 6e should be reexamined one by one to see if they are all indeed transcription factors.

Comment 2. In page 4, line 144, I think that (Fig. 3d) should be (Fig. 2d).

Reviewer #2 (Remarks to the Author):

The revised version has addressed my concerns about the paper and can be published on Communications Biology after professional embellishment with proper grammar, spelling, and composition.

Reviewer #1:

1) The authors list TFF3 as a transcription factor, but to my knowledge and my quick
Google and PubMed searches, TFF3 is considered to be a secreted protein but not
an established transcription factor. There is a possibility that I might have missed
important papers on TFF3 as a transcription factor though. But if my knowledge
and searches are correct, the authors need to rewrite a part of the text and reanalyze
their data, especially Figure 6. And I recommend all of the transcription factors that
the authors listed in Figure 6e should be reexamined one by one to see if they are
all indeed transcription factors.

**Response:** Thank you for your comments. We agree this is an important point, and
TFF3 indeed is a secreted protein traditionally. Few studies have shown that TFF3
is a transcription factor except for the study by Xie et al
(DOI:10.1093/nsr/nwaa180)¹. The authors indicated that CREM, TFF3 and NFIX
as newly identified transcription factors may regulate hematopoiesis in cooperating
with canonical networks¹. In addition, during the revision we have further analyzed
our data to reexamined all of the transcription factors. We identified transcription
factors that play regulatory roles in AC and MC using pySCENI
(10.1038/nmeth.4463; doi: 10.1038/s41596-020-0336-2). The pySCENI based on
the database cisTarget (<https://resources.aertslab.org/cistarget/>) which clearly states
TFF3 and others as transcription factors. All of the transcription factors that we
listed have been reexamined one by one and all of them can be find in the
transcription factor list. Finally, we thank this reviewer once again for the points
they raise.

1.Xie X, *et al.* Single-cell transcriptomic landscape of human blood cells. *Natl sci*
*rev* **8**, nwaa180 (2021).

2) In page 4, line 144, I think that (Fig. 3d) should be (Fig. 2d).

**Response:** Thank you for your comments. The error has been corrected.

Reviewer #2:

1) The revised version has addressed my concerns about the paper and can be

published on Communications Biology after professional embellishment with
proper grammar, spelling, and composition.

**Response:** Thank you for your comments. We have made corrections according to
the reviewer's comments. Furthermore, we have polished the manuscript to meet
the language standards of the journal. Modifications were highlighted in yellow.